# The micro-economic effects of COVID-19 containment measures: A simple model and evidence from China

**Wenxuan Chen[1], Songlei Chao[1]\*, Jianliang Ye[2]**

**1** Research Center for Intelligent Society and Governance, Research Institute of Interdisciplinary Innovation, Zhejiang Lab, Hangzhou, Zhejiang, China, **2** School of Economics, Zhejiang University, Hangzhou, Zhejiang, China

\* chaosonglei@zhejianglab.com

**Data Availability Statement:** All relevant data are available from the website: https://osf.io/5p8hy/.

**Funding:** This research was funded by Zhejiang Office of Philosophy and Social Science's Zhejiang Provincial Philosophy and Social Sciences Planning Project (www.zjskw.gov.cn), under Grant

## Abstract

Optimizing the trade-off between economic growth and public health is a major goal of public administration, especially during public health events. Although containment measures are widely used to combat the Covid-19 outbreak, it is still debated how the measures affect the economy. Using a simplified susceptible-infected-recovered (SIR) model, this study investigates the dynamic impact of lockdown policy on social costs during the epidemic and the underlying mechanism, revealing that the lockdown policy has both a "shutdown effect" and an "anti-epidemic effect", and should be implemented and lifted in a timely manner. Based on a micro-level dataset of 57,547 private enterprises in China in 2020, this study provided empirical evidence for the presence of negative "shutdown effect" and positive "anti-epidemic effect" of lockdown on reopening, both of which are in part mediated by labor input, factor mobility, and market demand recovery. Furthermore, the shutdown effect is weaker in regions with sufficient testing and quarantine resources, government capacity and preference for targeted response, whereas the anti-epidemic effect is stronger in densely populated areas with relatively low public compliance. Additionally, digital measures can aid in the containment of epidemics. The findings not only contribute to a better understanding of the rationality and effectiveness of the lockdown policy, but also provides practical evidence and implications for the government to improve the synergistic efficiency of epidemic control tools and strengthen the resilience of local economic growth.

## Introduction

The spread of COVID-19 has exacerbated the conflict between human health and economic well-being [1]. Containing COVID-19 while accelerating economic recovery is an urgent challenge for academics and policymakers. In contrast to most studies on the COVID-19 epidemic analysis that reveal the negative effects of the lockdown policy on the economy or public health [2–7], or the economic effects of the epidemic relief policies [8–11], the purpose of this work is to investigate the dynamic trade-off between current economic loss and future economic growth for the epidemic management strategy. Several studies have employed the Susceptible-

Agreement No. 23NDJC389YBM, and Zhejiang Provincial Big Data Development Administration's Zhejiang Digital Government Construction Theory Research Project, under Grant Agreement No. ZZCG2022F-CS-108. The funders had no role in the study design, data collection and analysis, decision to publish, or preparation of the manuscript.

**Competing interests:** The authors have declared that no competing interests exist.

Infected-Recovered (SIR) model and numerical simulation to study the trade-off between health and economic aspects, as well as to frame the optimal features of the optimal strategy [12, 13]. Our approach combines a simplified SIR model with enterprise-level data to investigate whether and how the epidemic management strategy affects economic recovery while maintaining public health safety.

The Chinese government has implemented swift and effective containment measures [14], opening the way for economic development and the resumption of work and production. By August 2022, the number of cases worldwide had risen to 584 million, with 6.42 million deaths. Due to the epidemic, most countries' GDPs experienced negative growth in 2020, and total GDP fell from 84.7 trillion in 2019 to 81.9 trillion in 2020 (in constant US dollars in 2015). China has led the way in epidemic control and economic recovery. Its GDP increased in 2020. China's economic recovery, as the world's most populous country, has not come at the expense of public health. China's cumulative Covid-19 cases accounted for 1% of the world's total as of August 2022, and death cases accounted for less than 0.4%, both far below the proportion of its total population in the world. In particular, emergency response policies have played an important role in China's epidemic management strategy. Since Wuhan's delayed lockdown on January 23, 2020, Chinese governments at all levels have launched Level 1 emergency response, and many cities and communities have been locked down to prevent the spread of COVID-19. China won the first battle against COVID-19 in just three months as a result of these stringent measures. According to the Emergency Response Law of the People's Republic of China and the National Emergency Response Plan for Public Emergencies, all provinces in China issued and dynamically adjusted their respective emergency response levels, as well as the local lockdown intensity. This makes it possible to study the dynamic economic impact of the lockdown.

Overall, compared to the existing literature on the effects of COVID-19 containment measures, the main findings of our work are four-fold:

- The lockdown policy, in theory, has both a "shutdown effect" and an "anti-epidemic effect," which is the inner rationale of the trade-off between current loss and future growth. The "shutdown impact" gauges the output cost brought on by the lockdown's restrictions on the non-infected individuals' activities, whereas the "anti-epidemic effect" gauges the long-term health and economic benefits brought on by the lockdown's suppression of new infections.

- Because of the inverted U-shaped pattern of the lockdown policy's effect, strict containment measures should be implemented early in the pandemic and the lockdown lifted in a timely manner. The anti-epidemic effect is large enough in the early stages of the epidemic to make the lockdown policy cost-effective. As the epidemic spreads due to a late or ineffective lockdown, the anti-epidemic effect diminishes and even reverses, and hence the lockdown raises social costs.

- The micro evidence from China demonstrates the existence of the "shutdown effect" as well as the "anti-epidemic effect". The interaction between the epidemic and containment measures affects the resumption of production through three channels: labor input, factor mobility, and market demand recovery.

- Moderation analysis reveals that the lockdown's shutdown effect can be effectively mitigated by local testing and quarantine resources, government resource mobilization capacity, and government's preference for scientific and targeted response, and that the anti-epidemic effect is stronger in densely populated areas, areas with better air quality, or areas with low citizen compliance. Additionally, lockdown measures can be partially substituted by digital measures.

## Materials and methods

### Theoretical framework and hypotheses

Based on the variation of the SIR model constructed by Alvarez et al. (2021) [12], agents can be divided into the susceptible $S_t$, the infected $I_t$, and the recovered $R_t$; that is,

$$N_t = S_t + I_t + R_t \qquad \text{for all } t \geq 0.$$

A rate $\phi(I_t)$ per unit of time of those infected die. The population decreases due to death according to

$$\dot{N}_t = -\phi(I_t)I_t$$

In the typical SIR model, $R_t$ represents the removed, including both the cured that are assumed to be immune and the dead. However, $R_t$ represents the cured only herein, and the dead is reflected in the decrease in the total population. The government can decide to lockdown a fraction $L_t \in [1,1)$ of those susceptible and those infected. Recovered agents need not be locked down. As assumed, the lockdown can curb virus transmission effectively with probability $\theta < 1$, and hence, $(1 - \theta L_t)$ agents can still transmit the virus. When the virus is transmitted from the infected to the susceptible, the susceptible becomes infected. The law of motion of the susceptible agents then is

$$\dot{S}_t = -\beta S_t(1 - \theta L_t)I_t(1 - \theta L_t),$$

where $\beta$ is the number of susceptible agents infected by each contact between the infected and the susceptible, i.e., the basic reproduction number. Given the recovery rate $\gamma$ (cure rate + fatality rate), the law of motion for the number of the infected is

$$\dot{I}_t = \beta S_t(1 - \theta L_t)I_t(1 - \theta L_t) - \gamma I_t.$$

Measuring the lockdown effectiveness, $\theta$ quadratically reduces the number of contacts between the infected and the susceptible during lockdown, and further reduces new infections.

### Planner's objectives

Alvarez et al. (2021) [12] also assumed that each agent alive and not in lockdown produces $w$ units of output, and that agents survive unless they die from the infection. The time discount rate is $r$ and with probability $v$ per unit of time vaccines and cures simultaneously appear, i.e., all are cured or immune. The planner aims at minimizing the social costs due to the lockdown policy: $min \int_0^\infty e^{-(r+v)t} \left( wL_t(S_t + I_t) + I_t\phi(I_t)\left[\frac{w}{r} + x\right] \right) dt$

The problem is a tradeoff between the output costs of lockdown and the fatality costs which is the product of the number of deaths per period times the shadow value assigned to each death, or the value of a statistical life (VSL) [15]. The VSL here measures the present value of the loss of direct income ($w/r$) of the dead and the extra cost ($x$) on individual and family well-being such as suffering. However, due to limited testing and quarantine resources or a one-size-fits-all lockdown policy, the recovered cannot be identified and released from lockdown in time. The parameter $\tau$ measures the probability of a test being available. The planner's problem is modified as:

$$\min_L \int_0^\infty e^{-(r+v)t} \left( wL_t[\tau(S_t + I_t) + (1 - \tau)(S_t + I_t + R_t)] + I_t\phi(I_t)\left[\frac{w}{r} + x\right] \right) dt$$

If there are sufficient testing and quarantine resources ($\tau = 1$), the recovered agents will be released from lockdown after quarantine. In this case, only the susceptible and infected agents would be in lockdown, which is called selective or targeted epidemic control. Without quarantine ($\tau = 0$), the entire labor force would be locked down, which is called blanket or one-size-fits-all lockdown.

This model lacks heterogeneity in fatality rates and diffusion rates. It represents the social costs resulting from a lockdown policy that cannot be distinguished among different types of agents, which aligns with the policy implemented in prefecture-level cities in China during the early stages of the outbreak. See Mena et al. (2021) [16] for a comprehensive analysis of the strong association between socioeconomic status and COVID-19 outcomes, and Acemoglu et al. (2020) [17] for an extension that allows for group-specific lockdowns. However, this model incorporates the effectiveness of lockdown measures as well as the capabilities of testing and quarantine to analyze heterogeneity at the regional level.

The planner solves the Hamilton-Bellman-Jacobi equation [18]:

$$(r + v)V(S, I) = \min_L \{\omega L[\tau(S + I) + (1 - \tau)(S_t + I_t + R_t)] + I\phi\left[\frac{w}{r} + x\right] - V_s$$
$$\cdot \beta SI(1 - \theta L)^2 + V_I(\beta SI(1 - \theta L)^2 - \gamma I)\} \tag{1}$$

$V(S,I)$ is interpreted as the minimum expected discounted cost of implementing the lockdown policy. Given the number of susceptible and infected agents in period $t$, the social costs of lockdown include the output costs and fatality costs in period $t$, referred to as "current costs", and the changes in social costs caused by the decrease in the susceptible population and the increase in the infected population due to new infections ($\beta SI(1 - \theta L)^2$) in period $t$, referred to as "long-term costs".

Using the envelope theorem, we solve for the partial derivative of the value function with respect to the lockdown level $L$ in period $t$:

$$(r + v)\frac{\partial V(S, I)}{\partial L} = w[\tau(S + I) + (1 - \tau)(S + I + R)] + 2\theta(1 - \theta L)\beta SI(V_s - V_I)$$
$$= \underbrace{w[S + I + (1 - \tau)R]}_{\text{Shutdown effect}} + \underbrace{2\theta(1 - \theta L)\beta SI(V_s - V_I)}_{\text{Anti-epidemic effect}} \tag{1A}$$

First, enhanced lockdown "directly" leads to the stoppage of work and production, resulting in a current marginal cost of $w[S + I + (1 - \tau)R] > 0$, which is referred to as the "shutdown effect". In theory, recovered agents need not be locked down and can return to work immediately. However, the time for them to return to work depends on local testing and quarantine resources, and the government's ability and attitude toward epidemic control (one-size-fits-all vs. targeted; length of quarantine). The shutdown effect increases with the proportion of the recovered, $R$, and decreases with the quarantine effectiveness, $\tau$, and thus decreases with the interaction between the recovered and testing and quarantine resources. The higher the proportion of the recovered, the more immune or low-risk individuals will be in lockdown due to the lack of testing and quarantine resources or overly aggressive measures adopted by local governments to achieve the epidemic control goals, and the greater the output costs.

**Hypothesis 1a:** In the short term, the lockdown inhibits the resumption of work and production, and a high cure rate exacerbates this negative effect.

**Hypothesis 1b:** In the short term, the lockdown inhibits the resumption of work and production, and the lack of testing and quarantine resources or the government's aggressive containment strategy exacerbates this negative effect.

Second, enhanced lockdown mitigates virus transmission from infected agents, inhibits new infections, thereby producing a long-term marginal impact $2\theta(1 - \theta L)\beta SI(V_S - V_I)$, which is referred to as the "anti-epidemic effect". $2\theta(1 - \theta L)\beta SI$ is the number of new infections that can be averted by marginal changes in the lockdown level, increasing at the early stage and decreasing later. $(V_S - V_I)$ measures the marginal social costs including marginal output cost and marginal fatality cost caused by a new infection, and its sign and magnitude are related to the epidemic stage. The social cost $V(S, I)$ is a nonlinear function of the number of the susceptible and infected agents, but its value range has boundaries [12]. When $I = 0$, i.e., there are no COVID-19 cases, $V(S,0) = 0$. As the number of infected agents grows from zero (the number of susceptible agents decreases), the social cost rises rapidly. Therefore, it is not difficult to conclude that in the early stage of the epidemic, $V_I > 0$ and $V_S < 0$, that is, the anti-epidemic effect is positive and increases with the number of the infected agents, and the current lockdown reduces future social costs by reducing new infections. When the susceptible population tends to zero, i.e., the whole population tends to be infected and immune, the social costs of lockdown will decrease to the boundary value $V(0,I) = \phi I(w/r+x)/(r+v+\gamma)$, i.e., the discounted value of economic cost due to the death of infected agents. An increase in the number of susceptible agents (a decrease in the number of infected agents) will result in additional output costs from the lockdown, thereby increasing the social costs. Therefore, at the end of the epidemic, $V_S > 0$ and $V_I < 0$, that is, the anti-epidemic effect is negative. In a nutshell, the anti-epidemic effect is in an inverted U-shaped pattern. The lockdown reduces long-term social costs during the early stages of the epidemic, but the continued increase in infections will eventually erode the positive anti-epidemic effect. In the occasion of widespread infection, the lockdown will raise social costs. This explains why some countries have implemented herd immunity strategies. From the standpoint of economic recovery, the planner should increase the lockdown level early in the epidemic and gradually lift the lockdown later. This finding is also consistent with the numerical simulation results for the socially optimal equilibrium [12, 13].

**Hypothesis 2a:** In the early stage of the epidemic, the lockdown helps prevent new infections, and the increase in the number of infected agents strengthens the anti-epidemic effect. In the later stage of the epidemic, a further increase in the number of infected agents may weaken the anti-epidemic effect.

Moreover, the anti-epidemic effect of lockdown is moderated by the number of the susceptible, the basic reproduction number, and the lockdown effectiveness. The number of new infections $2\theta(1 - \theta L)\beta SI$ that can be averted by a marginal increase in the lockdown level depends mainly on the transmission rate and lockdown effectiveness. With a moderate lockdown effectiveness, the higher the transmission rate ($\beta SI$), the stronger the anti-epidemic effect of the lockdown. Specifically, assuming that the novel coronavirus does not mutate in the early stage of the epidemic and the basic reproduction number $\beta$ remains unchanged, the denser the population, the more susceptible agents the infected agents may come into contact with, the more new infections are averted by the lockdown, and the stronger the anti-epidemic effect. Therefore, widespread lockdown should be implemented in areas with widespread infection and high population density. Given the population density, when the lockdown effectiveness is extremely low (e.g., in home monitoring with low compliance), even if the lockdown is implemented, it is still difficult to prevent virus transmission, and hence the anti-epidemic effect is limited. When the lockdown is extremely effective (e.g., in centralized quarantine or strictly supervised home quarantine), minimal new infections occur during the lockdown, and accordingly the anti-epidemic effect is marginally decreasing. It implies that the lockdown effectiveness achieved by home quarantine with high compliance may bring about a strong anti-epidemic effect.

**Hypothesis 2b:** In the early stage of the epidemic, the virus transmission rate and moderate lockdown effectiveness strengthen the anti-epidemic effect of the lockdown.

## Data description

The construction of a proxy variable for the degree of the lockdown is the foundation of this research. According to the *Emergency Response Law of the People's Republic of China* and the *National Emergency Response Plan for Public Emergencies*, emergencies are divided into four categories: natural disasters, accident calamity, public health emergencies, and public security emergencies. COVID-19 is a public health emergency. According to the nature, severity, controllability, and influence scope, emergencies are classified into four levels: Level 1 (extremely serious), Level 2 (serious), Level 3 (moderately serious), and Level 4 (general). A Level 1 emergency response is organized by the State Council and implemented in each province by the provincial governments under the leadership and command of the State Council. Downgrading the response level indicates a decrease in the scope, nature, and harm of the epidemic and entails that the response will be organized and implemented by lower-level governments. Levels 2, 3, and 4 responses are led and directed by the provincial, municipal, and county governments within their administrative regions, respectively. On February 24, 2020, at the press conference of the Joint Prevention and Control Mechanism of the State Council, Feng Mi, spokesperson and deputy director of the Publicity Department of the National Health Commission of the People's Republic of China, said, "*We have made progress in controlling COVID-19 at the current stage. A positive trend has been shown at the national level. Each province should adjust the emergency response level according to the local situation, so as to achieve precise COVID-19 prevention by zoning and grading and restore normal production and life*". On May 13, 2020, Qinghua He, a first-level inspector of the National Bureau of Disease Control and Prevention of the National Health Commission, said that the epidemic management work in China had entered a routine prevention and control state from an emergency state. It is clear that lowering the response level means not only transferring authority to lower-level governments, but also reducing the scope and intensity of epidemic control. It may be reflected, for example, in the length of home or centralized observation, inter-provincial (inter-city) mobility and community (village) access, transportation and tourism, and the return to school, work, and production. As a result, the emergency response level can be used as a proxy variable for the lockdown level.

All 31 provinces, municipalities and autonomous regions in mainland China, except for Tibet, successfully launched Level 1 emergency response from January 23 to 26, 2020. (Tibet launched Level 2 emergency response on January 27, 2020, and did not officially adjust or cancel the response level after that. However, it had recorded only 1 COVID-19 case, and was excluded from the analysis in this paper.) Downgrading from Level 1 to Level 2 or 3 response occurred at a different date in each area. For example, the Tibet Autonomous Region, which was the least hit by COVID-19, adjusted its emergency response to Level 2 as early as January 27, 2020. Hubei, the hardest-hit province, didn't downgrade its emergency response to Level 2 until May 2, 2020. Beijing, as well as nearby Tianjin and Hebei, had the longest duration of Level 1 response and implemented relatively aggressive containment measures, which is related to the uniqueness of Beijing as the capital. Guangdong and Jiangsu, which are located in the developed eastern coastal areas, had the shortest duration of Level 1 response, which may be related to local economic considerations. As of the end of March 2020, no new confirmed cases had been reported in 14 provinces for more than 7 days, 21 provinces had reported zero confirmed cases, and 21 provinces had successively downgraded the emergency response to Level 3 or 4 (see **Table 1** and **Fig 1**) [19, 20]. According to **Table 1**, an ordered

**Table 1. Start date of emergency response level in each province/autonomous region/municipality ("/" denotes default).**

| Region | Level 1 | Level 2 | Level 3 | Level 4 | Region | Level 1 | Level 2 | Level 3 | Level 4 |
|---|---|---|---|---|---|---|---|---|---|
| Zhejiang | January 23, 2020 | March 2, 2020 | March 23, 2020 | / | Hebei | January 24, 2020 | April 30, 2020 | June 6, 2020 | / |
| Guangdong | January 23, 2020 | February 24, 2020 | May 9, 2020 | / | Jiangsu | January 15, 2020 | February 24, 2020 | March 27, 2020 | / |
| Hunan | January 23, 2020 | March 19, 2020 | March 31, 2020 | / | Hainan | January 15, 2020 | / | February 26, 2020 | / |
| Hubei | January 24, 2020 | May 2, 2020 | June 13, 2020 | / | Xinjiang | January 15, 2020 | February 25, 2020 | March 8, 2020 | March 21, 2020 |
| Anhui | January 24, 2020 | February 25, 2020 | March 15, 2020 | / | Henan | January 15, 2020 | March 19, 2020 | May 6, 2020 | / |
| Tianjin | January 24, 2020 | April 30, 2020 | June 6, 2020 | / | Heilongjiang | January 15, 2020 | March 4, 2020 | March 25, 2020 | / |
| Beijing | January 24, 2020 | April 30, 2020 | June 6, 2020 | / | Gansu | January 15, 2020 | / | February 21, 2020 | May 11, 2020 |
| Shanghai | January 24, 2020 | March 24, 2020 | May 9, 2020 | / | Liaoning | January 15, 2020 | / | February 22, 2020 | / |
| Chongqing | January 24, 2020 | March 19, 2020 | March 24, 2020 | / | Shanxi | January 15, 2020 | February 24, 2020 | March 19, 2020 | / |
| Sichuan | January 24, 2020 | February 26, 2020 | March 25, 2020 | / | Shaanxi | January 15, 2020 | / | February 28, 2020 | / |
| Jiangxi | January 24, 2020 | March 12, 2020 | March 20, 2020 | / | Inner Mongolia | January 15, 2020 | / | February 26, 2020 | / |
| Yunnan | January 24, 2020 | / | February 24, 2020 | / | Jilin | January 15, 2020 | February 26, 2020 | March 20, 2020 | / |
| Guizhou | January 24, 2020 | / | February 23, 2020 | / | Ningxia | January 15, 2020 | February 28, 2020 | May 6, 2020 | / |
| Shandong | January 24, 2020 | March 8, 2020 | May 6, 2020 | / | Qinghai | January 26, 2020 | / | February 26, 2020 | March 6, 2020 |
| Fujian | January 24, 2020 | / | March 19, 2020 | / | Tibet | / | January 27, 2020 | / | / |
| Guangxi | January 24, 2020 | / | February 24, 2020 | / | | | | | |

categorical variable for daily emergency response (Levels 1, 2, 3, and 4 are coded as 3, 2, 1, and 0, respectively) and hence a monthly average emergency response level were generated.

Three types of data are used in this study: (a) Data on COVID-19 cases: daily panel data of COVID-19 cases, as of June 2020, for prefecture-level cities in China were collected from the Chinese Center for Disease Control and Prevention and provincial health commissions. These sources reported the cumulative confirmed, dead, and recovered COVID-19 cases in each city on a daily basis. The number of new cases per day was calculated, and monthly data on cumulative and new cases was generated. (see **S1 Table**). As shown by **Fig 1** and **S1 Table**, at the end of March, COVID-19 was well under control at the national level. The average emergency response level in April has a clear positive correlation with the total number of cases at the end of March, but no clear relationship with the number of new cases in April, indicating that response is usually delayed and endogeneity is not a major concern. To ensure the reliability of our data, we cross-checked it with three other open data sources: https://www.tianditu.gov.cn/coronavirusmap, and https://github.com/BlankerL/DXY-COVID-19-Data, and https://github.com/CSSEGISandData/COVID-19 (operated by the Johns Hopkins University Center for Systems Science and Engineering). We found some inconsistencies primarily in the numbers reported for cities in Hubei province, where Wuhan is located. These numbers underwent

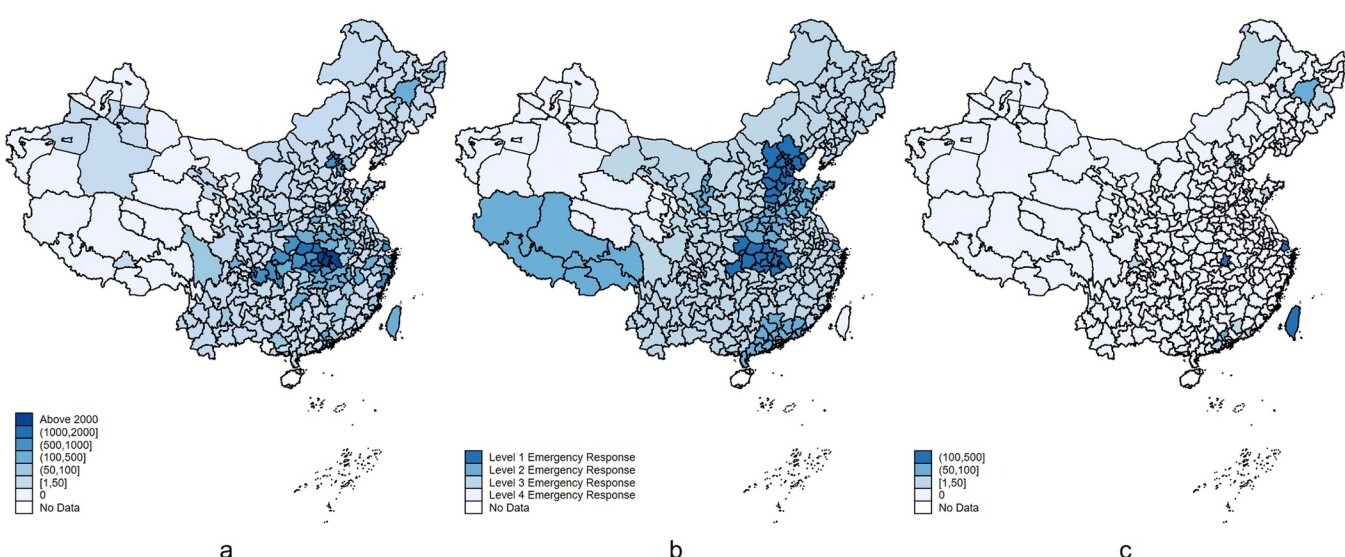

**Fig 1. Distribution of COVID-19 cases and emergency response level in each city in China.** (a) Regional distribution of cumulative confirmed cases at the end of March; (b) Regional distribution of average emergency response level in April; (c) Regional distribution of newly diagnosed cases in April. The map in Fig 1 is plotted with Stata, and the data sources can be available from: https://osf.io/5p8hy/.

significant adjustments officially following the Wuhan lockdown. After excluding Hubei province, we did not find any significant systematic bias between the different data sources. Besides, the reopening of Hubei province was overseen by the central government. Therefore, we made the decision to exclude any cities from Hubei province from our sample. **(b)** Data on the resumption of work and production reopening of enterprises: Through the private enterprise survey system of the All-China Federation of Industry and Commerce (ACFIC), the results of the second and third questionnaire surveys on private enterprise operation in China in 2020 began to be collected anonymously on May 26 and July 8, 2020, respectively, covering 337 prefecture-level cities (including autonomous prefectures and provincial districts and counties). The third questionnaire survey added and removed a large number of samples on the basis of the second questionnaire survey. To make full use of the data, the results of the two surveys were combined to create mixed cross-sectional data for analysis. Combining the results of two surveys not only increases data volume, but also reveals the dynamic impact of earlier containment measures on the work and production resumption after entering the routine COVID-19 management stage. A total of 27,040 and 30,507 enterprises were included in the two surveys, respectively, covering basic characteristics of the enterprises, such as location, industry, and size, as well as work/production resumption levels in May and June, and receipt of government and financial support. The work/production resumption level is a five-level ordered variable (1 = 0–10%, 2 = 10–30%, 3 = 30–50%, 4 = 50–80%, 5 = 80% or higher). Accordingly, we create two dummy variables, *resdummy* and *labdummy* (0 = 0–50%, 1 = 50% or higher). **(c)** Data on city characteristics: Data of the following characteristics of 296 prefecture-level cities in 2018 in China (excluding provincial counties and autonomous prefectures; data missing for Zunyi and Sansha) from *the China City Statistical Yearbook 2019* are included: average population, GDP per capita (10,000 yuan), medical resources (proportion of licensed physicians to the population and hospital size), government size (proportion of general public budget expenditure in GRP and proportion of employees in urban non-private units to the population), economic openness (dependence on foreign trade and level of external financing), informatization level (proportion of employees in the ICT industry to the population and

number of mobile phone subscribers per capita), air quality (PM2.5), postal service level, population density in built-up areas, and household size. The Weibo Government Affairs Index (ranking for April 2020) jointly issued by Sina Weibo and People's Daily is also included. Moreover, a number of city characteristics are included to measure the possible moderating factors of the shutdown and anti-epidemic effects, which ensures the robustness of the analysis.

Descriptive statistics of the data are shown in **Table 2**. There was no significant difference in the proportion of work or production resumption in May and June, but the resumption of work (82.9%) was significantly faster than that of production (66.3%). In fact, as COVID-19 was well controlled, enterprises gradually resumed work and reopened. However, the resumption of work does not mean the resumption of production, nor does it mean the resumption of business or sales. After the enterprises resumed work, they were still subject to various constraints, such as the work resumption rate of upstream suppliers, the supply of raw materials, downstream logistics efficiency, and demand recovery. Therefore, the epidemic and its containment measures have a far-reaching impact on the resumption of production.

In terms of sectors, the highest production resumption level (72.3%) was observed for enterprises in the secondary industry (data for June; the same below), and diversified enterprises had the lowest level (56.3%). In terms of industries, the hotel and restaurant industries had the lowest production resumption levels (57.7% and 65.1%), and the postal industry had the highest level (84.6%). In terms of firm size, large-sized enterprises had the highest production resumption level (79.5%), and micro-enterprises had the lowest level (54.6%). In terms of regions, Shanghai had the highest production resumption level (83.7%), and Liaoning Province had the lowest level (40.0%). Due to strict containment measures, Beijing had significantly slower production resumption (56.4%) than other first-tier cities, ranking sixth from the bottom. Hubei, the hardest-hit province, quickly resumed work and production after city lockdown, ranking seventh in the level of production resumption. Regarding the sense of gain and effectiveness of support policies, as of May, 81.4% of the interviewed enterprises enjoyed at least one government preferential policy on taxes and dues, and 50.9% received at least one type of financial support; as of June, the proportions of enterprises receiving at least one preferential policy on taxes and dues and one type of financial support increased to 92.6% and 68.0%, respectively. In particular, the exemption of pension, unemployment, and work-related injury insurance premiums paid by employers for micro, small, and medium-sized enterprises (22.7%) and interim deferral or appropriate refund of social insurance premiums (22.5%) were regarded as the most effective policies. Not blindly withdrawing, cutting off, or delaying issuing loans to enterprises was selected as the most effective financial support.

## Empirical model

To analyze the impact of containment measures (lockdown) on the total social cost, the theoretical model (1) was rewritten to construct an empirical model as follows:

$$Y_{idcp} = \beta_0 + \beta_1 rr_p + \delta_1 rcv_{ct} \times rr_p + \delta_2 cfm_{ct} \times rr_p + \boldsymbol{\beta_2 X_{idcp}} + \lambda_p + \varphi_d + \varepsilon_{idcp} \qquad (2)$$

$Y_{idcp}$ is the production resumption level of enterprise i in industry d in city c of province (autonomous region or municipality) p in May, with "whether production is basically resumed", *resdummy*, as the dependent variable. $rr_p$ is the average emergency response level of province *p* and used as a proxy variable for the city lockdown level. It is seen from Eq (1A) that the social cost (stoppage of work and production) is a nonlinear function of the lockdown, including both the shutdown and anti-epidemic effects. If the anti-epidemic effect exceeds the shutdown effect, the coefficient of the lockdown is positive, otherwise it is negative. $rcv_{ct}$ is the

**Table 2. Descriptive statistics of key variables.**

| Variable | Description | Sample size | Mean | Standard deviation | Min | Max |
|---|---|---|---|---|---|---|
| 1) Provincial emergency response level | | | | | | |
| rr_3m | Average emergency response level in March (Level 4 = 0, Level 3 = 1, Level 2 = 2, Level 1 = 3) | 26,900 | 1.790 | 0.710 | 0 | 3 |
| rr_m | Average emergency response level in April | 26,932 | 1.400 | 0.714 | 0 | 3 |
| rr_5m | Average emergency response level in May | 26,929 | 1.102 | 0.408 | 0 | 2.032 |
| 2) COVID-19 data in prefecture-level cities in China (excluding imported or unconfirmed cases) | | | | | | |
| cfm_3mt | Cumulative cases in March | 404 | 204.421 | 2500.537 | 0 | 50,006 |
| rcv_3mt | Cumulative recoveries in March | 404 | 188.693 | 2301.385 | 0 | 46,002 |
| dth_3mt | Cumulative deaths in March | 404 | 8.203 | 127.125 | 0 | 2548 |
| cfm_4mn | New cases in April | 404 | 4.423 | 31.914 | -1 | 384 |
| rcv_4mn | New recoveries in April | 404 | 6.393 | 47.939 | -1 | 718 |
| dth_4mn | New deaths in April | 404 | 3.290 | 65.721 | 0 | 1321 |
| 3) Enterprise characteristics in the second questionnaire survey | | | | | | |
| resratio | Production resumption level (1 = 0–10%, 2 = 10–30%, 3 = 30–50%, 4 = 50–80%, 5 = 80% or higher) | 25,788 | 3.797 | 1.303 | 1 | 5 |
| resdummy | Whether production is basically resumed (1 = more than 50% of production resumed, 0 = less than 50% of production resumed) | 25,788 | 0.664 | 0.472 | 0 | 1 |
| labret | Work resumption level (using the same definition as the resratio levels) | 25,780 | 4.515 | 0.944 | 1 | 5 |
| labdummy | Whether work is basically resumed (1 = more than 50% of work resumed, 0 = less than 50% of work resumed) | 25,780 | 0.879 | 0.326 | 0 | 1 |
| ordup | Whether the order volume is decreased (1 = the same or increased, 0 = decreased) | 25,090 | 0.578 | 0.494 | 0 | 1 |
| dmdexp2 | Whether the expected market size is decreased (1 = the same or increased, 0 = decreased) | 25,731 | 0.541 | 0.498 | 0 | 1 |
| foreign | Whether engaged in import and export (1 = yes, 0 = no) | 27,121 | 0.324 | 0.468 | 0 | 1 |
| capsat | Whether capital is adequate (1 = yes, 0 = no) | 27,121 | 0.388 | 0.487 | 0 | 1 |
| policy_a0 | Enjoying at least one COVID-19-related government preferential policy on taxes and dues | 27,121 | 0.814 | 0.389 | 0 | 1 |
| finsppt_a0 | Receiving at least one type of COVID-19-related financial support | 27,121 | 0.509 | 0.500 | 0 | 1 |
| 4) Enterprise characteristics in the third questionnaire survey | | | | | | |
| resratio | Production resumption level (1 = 0–10%, 2 = 10–30%, 3 = 30–50%, 4 = 50–80%, 5 = 80% or higher) | 30,505 | 3.667 | 1.571 | 0 | 5 |
| resdummy | Whether production is basically resumed (1 = more than 50% of production resumed, 0 = less than 50% of production resumed) | 30,505 | 0.663 | 0.473 | 0 | 1 |
| labret | Work resumption level (using the same definition as the resratio levels) | 30,505 | 4.242 | 1.454 | 0 | 5 |
| labdummy | Whether work is basically resumed (1 = more than 50% of work resumed, 0 = less than 50% of work resumed) | 30,505 | 0.829 | 0.376 | 0 | 1 |
| ordup | Whether the order volume is decreased (1 = the same or increased, 0 = decreased) | 25,090 | 0.578 | 0.494 | 0 | 1 |
| dmdexp2 | Whether the expected market size is decreased (1 = the same or increased, 0 = decreased) | 25,731 | 0.541 | 0.498 | 0 | 1 |
| foreign | Whether engaged in import and export (1 = yes, 0 = no) | 27,121 | 0.324 | 0.468 | 0 | 1 |
| capsat | Whether capital is adequate (1 = yes, 0 = no) | 27,121 | 0.388 | 0.487 | 0 | 1 |
| policy_a0 | Enjoying at least one COVID-19-related government preferential policy on taxes and dues | 30,600 | 0.926 | 0.262 | 0 | 1 |
| finsppt_a0 | Receiving at least one type of COVID-19-related financial support | 28,201 | 0.680 | 0.466 | 0 | 1 |
| 5) Basic city characteristics (moderating factors) | | | | | | |
| pdoctor | Proportion of licensed physicians to the population (per 10,000 people) | 291 | 26.326 | 12.169 | 10.996 | 88.308 |
| doc_hos | Hospital size (number of doctors/number of hospitals) | 296 | 112.260 | 48.131 | 4.000 | 310.615 |
| pbexp_grp | Proportion of general public budget expenditure in regional GRP (%) | 296 | 0.235 | 0.162 | 0.074 | 1.554 |
| urempratio | Proportion of employees in urban non-private units to the population (per 10,000 people) | 291 | 1322.576 | 1280.429 | 315.386 | 10,969.000 |
| openness | Foreign trade dependence (import and export volume/regional GDP) | 288 | 0.159 | 0.246 | 0.001 | 1.621 |

(*Continued*)

**Table 2.** (Continued)

| Variable | Description | Sample size | Mean | Standard deviation | Min | Max |
|---|---|---|---|---|---|---|
| fdi_grp | External financing level (FDI/GRP, USD/yuan) | 270 | 0.002 | 0.003 | 0.000 | 0.030 |
| pictworker | Proportion of employees in the ICT industry to the population (per 10,000 people) | 290 | 24.444 | 60.941 | 1.313 | 614.254 |
| pmobile | Number of mobile phone subscribers per capita (subscribers/person) | 292 | 1.180 | 0.777 | 0.111 | 8.566 |
| pm25 | Annual average concentration of inhalable particles ($\mu g/m^3$) | 243 | 40.502 | 13.503 | 11.000 | 110.000 |
| ppostal | Per capita postal service income (yuan/person) | 291 | 434.122 | 1095.238 | 9.375 | 10,689.890 |
| popden | Population density in built-up areas (10,000 people/square kilometer) | 282 | 1.861 | 11.426 | 0.254 | 192.857 |
| housesize | Family size (registered population/number of households, person/household) | 296 | 3.084 | 0.486 | 2.036 | 4.813 |

Note: Negative values of the number of new cases in panel 2) may be due to changes in quarantine results (e.g., retesting positive during quarantine) or inter-regional transfer of patients. Only 622/57,721 of the samples had negative numbers of cures and infections in April, which were all from Dalian.

number of new recoveries in city $c$ at the end of month t. $\delta_1$ estimates the interaction between the emergency response level and the number of new recoveries, which is used to measure the *shutdown effect*. Hypothesis 1a suggests that the *shutdown effect* of the lockdown mainly stems from the restricted movement of recovered or low-risk individuals. The negative estimate of $\delta_1$ indicates the presence of the shutdown effect $cfm_{ct}$ is the number of new confirmed cases in city $c$ at the end of month $t$. $\delta_2$ estimates the interaction between the emergency response level and the number of new positive cases, which is used to measure the *anti-epidemic effect*. Hypotheses 2a and 2b suggest that, from the perspective of the *anti-epidemic effect*, the lockdown reduces the social costs due to new infections by limiting exposure to the virus. The non-negative estimate of $\delta_2$ indicates the presence of the anti-epidemic effect. Two things are noteworthy here: 1)New infections appear in both the shutdown and the anti-epidemic effects. Due to their opposite directions, $\delta_2$ would systematically underestimate the anti-epidemic effect. While the shutdown effect is significantly present, a non-negative estimate of $\delta_2$ still demonstrates the presence of the anti-epidemic effect. 2) There is a concern regarding potential biases in the reported confirmed cases, which may arise due to population-level health-seeking behavior, surveillance capacity, and the presence of asymptomatic individuals across different regions (Lu et al., 2021; Mena et al., 2021) [16, 21]. However, the utilization of advanced smart digital technologies in China, such as health codes, itinerary codes, and location codes, has significantly enhanced the efficiency of COVID-19 detection and monitoring of individuals at risk. This was particularly advantageous during the period of March and April 2020 when the number of cases was considerably lower, thus mitigating the potential biases associated with newly confirmed cases in our sample.

The control variables $X_{idcp}$ include the work resumption level, firm size (large, medium, small, and micro), and receipt of COVID-19-related preferential policies and financial support (the most effective policy for enterprises among the 17 preferential policies and the most effective financial support for enterprises among the 12 types of financial support) of enterprise $i$ to control for other unobservable factors that affect the resumption of work and production of enterprises, as well as COVID-19 data (infections, recoveries, and deaths) to control for COVID-19-related emergencies, and characteristics of city c (population, per capita GDP). The model also controls for the provincial fixed effect $\lambda_p$ and the industry fixed effect $\varphi_d$.

The 11 COVID-19-related preferential policies that have been implemented by the Chinse government during the second questionnaire survey include (1) tax reductions for enterprises producing key supplies for COVID-19 control; (2) extended carryover period and deferral of tax payment for enterprises with annual losses in difficult industries that have been greatly

affected by COVID-19; (3) interest subsidies and tax preferences for small and medium-sized enterprises included in the list of key enterprises for COVID-19 control issued by the central government; (4) special bailout funds; (5) reduction and exemption of administrative fees; (6) reduction and exemption of property rents for state-owned business premises; (7) interim deferral or appropriate refund of social insurance premiums; (8) interim deferral of electricity, water, and gas payment for enterprises and non-stop supply during the deferred payment period; (9) allowing enterprises to extend the deadline for tax declaration; (10) employment, job retention, and employee training subsidies for eligible enterprises; and (11) financing incentives. In the third questionnaire survey, additional policies were included: (12) loan repayment deferral or negotiation regarding deferral for small and micro enterprises and enterprises in difficulty; (13) exemption of pension, unemployment, and work-related injury insurance premiums paid by employers for micro, small, and medium-sized enterprises; (14) reduced industrial and commercial electricity, broadband, and private line rates; (15) exemption from value-added tax on services such as public transportation, restaurants and hotels, tourism and entertainment, and culture and sports; (16) deferred income tax payment for all small and micro enterprises and individual businesses; and (17) other policies.

The 12 types of financial support include (1) not blindly withdrawing, cutting off, or delaying issuing loans to enterprises; (2) extending or renewing loans for enterprises seriously affected by COVID-19; (3) appropriately reducing loan interest rates and increasing medium and long-term loans; (4) policy-based special loans; (5) low-interest credit loans; (6) bank credit loans; (7) government-subsidized loans; (8) relaxation of capital market business conditions; (9) financing through supply chain finance, commercial factoring, accounts receivable mortgage and pledge, intellectual property pledge, and other methods; (10) counter-guarantee requirements removed by government financing guarantee institutions, and re-guarantee fee halved by the National Financing Guarantee Fund; (11) government on-lending for emergency working capital; and (12) government bailout funds.

### Moderation analysis

As shown in Eq (1A), the difference in the moderating factors of the *shutdown* and *anti-epidemic effects* provides a theoretical basis for further verifying that containment measures have two opposite effects on the production resumption of enterprises. As stated in Hypotheses 1b and 2b, regional testing and quarantine resources and government attitude towards epidemic control (*moderator_1$_c$*) may moderate the *shutdown effect* of the lockdown, whereas population density and lockdown effectiveness (*moderator_2$_c$*) may moderate the *anti-epidemic effect*. Hence, the triple interaction model 2a is constructed as:

$$Y_{idcp} = \beta_0 + \beta_1 rr_p + \delta_1 rr_p \times rcv_{ct} + +\delta_2 rr_p \times cfm_{ct} + \delta_3 rr_p \times rcv_{ct} \times moderater\_1_c + \delta_4 rr_p$$
$$\times cfm_{ct} \times moderater\_2_c + \beta_2 X_{idcp} + \lambda_p + \varphi_d + \varepsilon_{idcp} \qquad (2A)$$

A variety of city characteristics are included to assess the potential moderating factors of the shutdown and anti-epidemic effects, ensuring the analysis's robustness. Table 3 briefly reports the selection of moderators. Regarding the *shutdown effect*, the proportion of licensed physicians to the urban population is used to measure the quantity of testing and quarantine resources, the average size of hospitals (number of doctors/number of hospitals) to measure the quality of testing and quarantine resources. We recognize that the suitability of these measures may vary in different contexts due to physical constraints, such as supply chain issues, impacting the availability of testing kits and essential devices across regions. However, China's well-developed transportation network has reduced the logistics disparity across prefecture-level cities, and, more importantly, the supply of kits/devices was mostly guaranteed as the

**Table 3. Moderating factors.**

| Moderating factors of the shutdown effect | | | Moderating factors of the anti-epidemic effect | | |
|---|---|---|---|---|---|
| **Selection basis** | **Moderating variable** | **Moderating effect** | **Selection basis** | **Moderating variable** | **Moderating effect** |
| Testing and quarantine resources | Proportion of licensed physicians | Reduction | Lockdown effectiveness 1 Digitalization level | Proportion of employees in the ICT industry | Inverted U |
| | Average hospital size | Reduction | | Number of mobile phone subscribers per capita | Inverted U |
| Testing and quarantine resource mobilization | Proportion of government expenditure | Reduction | | Weibo Government Affairs Index | Inverted U |
| | Proportion of employees in urban non-private units | Reduction | Lockdown effectiveness 2 External environment | PM2.5 concentration | Inverted U |
| Government attitude towards epidemic control | Foreign trade dependence | Reduction | | Per capita postal service income | Inverted U |
| | External financing level | Reduction | Transmission rate | Population density | Enhancement |
| | | | | Household size | Enhancement |

COVID-19 was under control during the sample period. Therefore, medical conditions, including the availability of medical services and designated hospitals for testing and treatment, serve as better predictors of testing and quarantine resources. Improvements in the quantity and quality of testing and quarantine resources may either reduce the shutdown effect by shortening the time it takes to liberate recovered individuals, or increase it if the local government adopts aggressive containment measures due to abundant testing and quarantine resources. To further distinguish between these two outcomes, the proportion of government general public budget expenditure in Gross Regional Product (GRP) and the proportion of employees in urban non-private units are used to assess the government's ability to mobilize resources for epidemic containment in terms of economic and staff size, respectively. If local governments with adequate testing and quarantine resources adopt aggressive containment measures or enforce a one-size-fits-all approach that restricts factors excessively and prolongs centralized isolation or quarantine for infected and susceptible individuals due to limited resource mobilization capacity, it may result in an intensified shutdown effect. In addition, foreign trade dependence and external financing level are used as proxy variables for government attitude towards epidemic control. The more dependent a region is on foreign trade, the greater economic cost it will suffer from the lockdown, and the less likely the government will adopt an aggressive one-size-fits-all containment strategy, which will help reduce the *shutdown effect*. $\delta_3$ estimates the effect of the moderators on the *shutdown effect*.

The moderating factors of the *anti-epidemic effect* include the lockdown effectiveness and the transmission rate of the virus. The proportion of employees in the ICT industry and number of mobile phone subscribers per capita are used as proxy variables for regional digitalization level. On the one hand, digitalization helps to enable working or studying from home to improve the lockdown effectiveness. On the other hand, local digitalization helps the horizontal transmission of epidemic information, which may enhance public compliance with containment measures [22], thus improving the lockdown effectiveness. The Weibo Government Affairs Index is used as a proxy variable for local government digitization. This index covers four dimensions, including communication, interaction, service, and recognition, and partially reflects the government's horizontal public opinion control. Research shows that the release of information related to the epidemic and its prevention policies through the government affairs microblog platform helps enhance public compliance [23]. The government affairs microblogs in April 2020 mainly focused on COVID-19 cases and related scientific knowledge. The Weibo Government Affairs Index reflects regional public awareness and cooperation with

containment measures and thus can be used as a proxy variable for lockdown effectiveness. In addition to the digitalization level, environmental factors also affect public attitudes towards staying at home. PM2.5 concentration was negatively correlated with outdoor activity tendency, such as visit rate of restaurants and shopping areas and school and work attendance [24] (Liu and Salvo, 2018). Residents were more likely to obey orders to stay at home in areas with poor air quality. Similarly, the more developed the express delivery industry, the higher the compliance with staying at home. Hence, both indicators can be used as proxy variables for lockdown effectiveness. Moreover, population density in built-up areas reflects regional overall population density, and average household size reflects the population concentration at the micro level. Both can be used as proxy variables for the transmission rate. $\delta_4$ estimates the effect of the moderators on the *anti-epidemic effect*.

## Results

### Shutdown and anti-epidemic effects

**Table 4** reports the regression results of the mixed cross-section data for empirical model (2). **Table 5** reports the regression results of the subsample data by period. The results can be summarized as follows: 1) On average, the lockdown significantly hindered the resumption of work and production, and had a greater negative impact on the return to work. 2) New infections in April impeded the resumption of work and production, while new recoveries (including recoveries and deaths) aided in the resumption of production. 3) In terms of regional characteristics, the increase in regional economic level, total population and population density, and the vertical administrative power of the government (proportion of employees in urban non-private units to the population) contributed to the resumption of work and production during the epidemic, whereas the government's horizontal public opinion control (Weibo Government Affairs Index) inhibited the resumption of work and production. Employees in urban non-private units are all persons who work in state organs, political party organs, social organizations, non-private enterprises, and public institutions at all levels and receive wages or other forms of remuneration. It mainly reflects the number of "in-system" staff. During the epidemic, the Chinese government deployed containment actions from top to bottom, and the "in-system" staff were the main actors to implement the containment strategy. In terms of enterprise characteristics, enterprises with import and export activities suffered short-term losses but recovered in June, and capital inadequacy is another factor that has directly hampered the resumption of work and production.

When the interaction terms between the number of new recoveries/infections and the lockdown are included, the direct hindering effect of the lockdown on the resumption of work and production is reduced (as shown by comparing Model 1 with Models 2, 3 and 4 in **Table 4**), and the coefficient of the direct effect on the resumption of production is almost halved. It indicates that output costs from containment measures such as lockdowns largely depend on the number of local cases. Specifically, the interaction between the number of recoveries and the lockdown is negative. The *shutdown effect* is stronger in regions with more recoveries. The interaction between the number of infections and the lockdown is positive. The anti-epidemic effect that promotes economic recovery is stronger in regions with more infections. **Table 5** shows that these two effects were always significant for the resumption of production, and significant for the resumption of work in June (the third questionnaire survey). As stated in Hypotheses 1 and 2, given the lockdown level, an increase in the number of recoveries in a region means a greater number of confined healthy people and greater output costs. On the other hand, the increase in the number of infections in a region means that the lockdown would reduce new infections by preventing virus transmission, thereby reducing output costs in the future.

**Table 4. Logit regression of work and production resumption on lockdown and local COVID-19 cases (mixed cross-section sample).**

| Model | 1 | 2 | 3 | 4 | 5 |
|---|---|---|---|---|---|
| | labdummy | labdummy | resdummy | resdummy | resdummy |
| main | | | | | |
| rr_m | -0.646*** | -0.599** | -0.415*** | -0.252** | -0.174 |
| | (0.240) | (0.240) | (0.144) | (0.112) | (0.171) |
| rcv_4mn | 0.0168 | 0.192 | 0.0271 | 0.321*** | 0.217*** |
| | (0.0258) | (0.155) | (0.0291) | (0.0493) | (0.0585) |
| cfm_4mn | -0.0436 | -0.225 | -0.0634 | -0.565*** | -0.346*** |
| | (0.0530) | (0.315) | (0.0591) | (0.0977) | (0.114) |
| Shutdown effect | | -0.113 | | -0.167*** | -0.112*** |
| | | (0.0744) | | (0.0292) | (0.0360) |
| Anti-epidemic effect | | 0.142 | | 0.292*** | 0.177** |
| | | (0.154) | | (0.0611) | (0.0736) |
| labret | | | | | 1.412*** |
| | | | | | (0.0307) |
| dth_4mn | 1.529*** | 0.844** | 1.464*** | 0.801*** | 0.765* |
| | (0.487) | (0.410) | (0.393) | (0.309) | (0.411) |
| pgrp | 0.0366** | 0.0392*** | 0.0168* | 0.0182* | 0.00807 |
| | (0.0151) | (0.0148) | (0.0100) | (0.00987) | (0.00926) |
| pop | 0.000537*** | 0.000523*** | 0.000355*** | 0.000321*** | 0.000216* |
| | (0.000171) | (0.000175) | (0.000121) | (0.000124) | (0.000113) |
| popden | 0.00511*** | 0.00539*** | 0.00733*** | 0.00752*** | 0.00728*** |
| | (0.000991) | (0.000981) | (0.000848) | (0.000858) | (0.000848) |
| wbscore | -0.0105** | -0.0117** | -0.00801** | -0.00926*** | -0.00647** |
| | (0.00514) | (0.00514) | (0.00340) | (0.00334) | (0.00284) |
| urempratio | 0.000104** | 0.000105** | 0.0000642** | 0.0000681** | 0.0000222 |
| | (0.0000515) | (0.0000527) | (0.0000324) | (0.0000333) | (0.0000349) |
| 1.foreign | -0.432*** | -0.431*** | -0.177*** | -0.175*** | -0.0189 |
| | (0.0663) | (0.0663) | (0.0422) | (0.0421) | (0.0369) |
| 1.capsat | 0.227*** | 0.226*** | 0.278*** | 0.278*** | 0.218*** |
| | (0.0653) | (0.0654) | (0.0412) | (0.0413) | (0.0385) |
| Other control variables | √ | √ | √ | √ | √ |
| Industry fixed effect | √ | √ | √ | √ | √ |
| Province fixed effect | √ | √ | √ | √ | √ |
| Month fixed effect | √ | √ | √ | √ | √ |
| pseudo $R^2$ | 0.100 | 0.100 | 0.085 | 0.086 | 0.247 |
| N | 44,061 | 44,061 | 44,072 | 44,072 | 44,072 |

Note: The sample is the mixed cross-sectional data of enterprises interviewed in the second and third questionnaire surveys. The dependent variables are dummies for whether production and work are basically resumed. The independent variables are the average emergency response level and number of recoveries and infections in each city in April. Models 2 and 4 include the interaction between COVID-19 cases (number of recoveries and infections in April) and lockdown in addition to the variables in Models 1 and 3, respectively (to estimate the shutdown and anti-epidemic effects, respectively). Model 5 includes the work resumption level in addition to the variables in Model 2. Other control variables include number of new deaths, regional per capita GRP, population and population density, Weibo Government Affairs Index (government's vertical information system), regional characteristics such as proportion of employees in urban non-private units, dummy for whether enterprises engaged in import and export, and dummy for whether enterprises had a funding gap to fill. Numbers in brackets are standard errors clustered at the city level.

* $p < 0.1$

** $p < 0.05$

*** $p < 0.01$.

**Table 5. Logit regression of work and production resumption on lockdown and local COVID-19 cases (sub-samples).**

| Model | The second questionnaire survey (May) | | The third questionnaire survey (June) | |
|---|---|---|---|---|
| | **labdummy** | **resdummy** | **labdummy** | **resdummy** |
| main | | | | |
| rr_m | -0.979*** | -0.189 | -0.0253 | -0.290 |
| | (0.144) | (0.188) | (0.385) | (0.187) |
| rcv_4mn | 0.0449 | 0.165** | 0.593*** | 0.254*** |
| | (0.0381) | (0.0645) | (0.0875) | (0.0884) |
| cfm_4mn | -0.0157 | -0.260** | -0.974*** | -0.426** |
| | (0.0706) | (0.126) | (0.161) | (0.169) |
| Shutdown effect | -0.0202 | -0.0947** | -0.330*** | -0.129** |
| | (0.0281) | (0.0400) | (0.0425) | (0.0539) |
| Anti-epidemic effect | -0.00175 | 0.152* | 0.553*** | 0.215** |
| | (0.0594) | (0.0819) | (0.0895) | (0.107) |
| labret | | 1.355*** | | 1.476*** |
| | | (0.0369) | | (0.0389) |
| 1.foreign | -0.456*** | -0.0725 | -0.345** | 0.113** |
| | (0.0759) | (0.0508) | (0.136) | (0.0527) |
| 1.capsat | 0.133** | 0.180*** | 0.297*** | 0.262*** |
| | (0.0612) | (0.0552) | (0.102) | (0.0441) |
| Other control variables | √ | √ | √ | √ |
| Industry fixed effect | √ | √ | √ | √ |
| Province fixed effect | √ | √ | √ | √ |
| Month fixed effect | √ | √ | √ | √ |
| pseudo $R^2$ | 0.098 | 0.237 | 0.114 | 0.258 |
| N | 21,189 | 21,192 | 22,872 | 22,872 |

Note: The samples are the cross-sectional data of enterprises interviewed in the second and third questionnaire surveys, respectively. The dependent variables are dummies for whether production and work are basically resumed. The independent variables are the average emergency response level and number of recoveries and infections in each city in April. Models 2 and 4 include the interaction between COVID-19 cases (number of recoveries and infections in April) and lockdown in addition to the variables in Models 1 and 3, respectively (to estimate the shutdown and anti-epidemic effects, respectively). Model 5 includes the work resumption level in addition to the variables in Model 2. Other control variables include number of new deaths, regional per capita GRP, population and population density, Weibo Government Affairs Index (government's vertical information system), regional characteristics such as proportion of employees in urban non-private units, dummy for whether enterprises engaged in import and export, and dummy for whether enterprises had a funding gap to fill. Numbers in brackets are standard errors clustered at the city level.

* $p < 0.1$

** $p < 0.05$

*** $p < 0.01$.

It should also be noted that COVID-19 and its containment policies had a far greater impact on production resumption than on work resumption. As previously stated, resumption of work does not imply resumption of production or sales. Resumption of production is always significantly more difficult than resumption of work. This is because, even after workers returned, businesses were still subject to various constraints, including the work resumption rate of upstream suppliers, raw materials supply, downstream logistics efficiency, and demand recovery. In other words, an enterprise's resumption of production is determined not only by its own work resumption status, but also by the work resumption levels of all enterprises in the supply chain and final market demand. The latter two aspects are also influenced by the epidemic and containment measures in the region where the enterprise is located. Model 5 in

Table 4 includes the work resumption level, *labret*, in addition to the variables in Model 4. Three findings are noteworthy: First, the resumption of work had a positive effect on the resumption of production, and the explanatory power of the model has been significantly improved (pseudo $R^2$ increased from 0.086 to 0.245), indicating that the work resumption level is a necessary control variable. Second, the resumption of work completely mediates the direct impact of the lockdown policy on the resumption of production. With zero cases of COVID-19, the lockdown hinders the resumption of production entirely because it restricts the return of employees to work. Third, when the work resumption level is controlled, the *shutdown* and *anti-epidemic effects* remain significant but decrease in magnitude. That is to say, the resumption of production is not only affected by the resumption of work, but also by the interaction between the epidemic and containment measures through other channels, such as the supply chain and final demand.

From the perspective of the supply chain, containment measures may have a cross-regional impact on the resumption of production. Under the lockdown, what kind of "spillover" the changes in the number of COVID-19 cases in other regions have on local enterprises is an empirical question. This will also directly determine whether the effect calculated by the base-line model is overestimated. To answer this question, we replaced the number of new local infections and recoveries with the number of new infections and recoveries in other regions within the same province. As shown in Table 6, three findings are noteworthy. First, when the number of COVID-19 cases in other regions is taken into account, the direct impact of lock-down on work and production resumption is no longer significant. Second, new infections and recoveries in other regions have coefficient signs that are opposed to those of new local infections and recoveries, and have a significant impact on production resumption. That is, the worsening of the epidemic in other regions will accelerate the resumption of production of local enterprises. The possible explanation is that, in the absence of a lockdown and barriers to the flow of people, resource elements and demand will flow to regions where the epidemic is better controlled, thereby promoting the resumption of production in these regions. It indicates that there is a "Covid-19 control competition" between regions. Third, the interaction between the number of new recoveries in other regions and the lockdown, i.e. *cross-regional shutdown effect* has a positive sign, while the *cross-regional anti-epidemic effect*, interaction between new infections in other regions and the lockdown, has a negative sign. Given the lock-down level, the more severe the epidemic (i.e., more new infections and fewer new recoveries) in other regions within the same province in the early stage of the epidemic, the more difficult it would be for local enterprises to resume production later. From another point of view, the *cross-regional shutdown and anti-epidemic effects* have signs that are opposite to those of local ones, also indicating the presence of a COVID-19 control competition between regions. The lockdown hindered inter-regional movement of other production factors and inter-regional logistics, thus reducing the competition between regions. This finding suggests that the *shutdown and anti-epidemic effects* estimated based on local COVID-19 cases are actually underestimated.

From the perspective of final demand, the slower resumption of production than of work could be attributed to an insufficient boost in market demand. The change in the number of enterprises' orders (1 = stay the same or increase, 0 = decrease) and their subjective expectation of market demand (1 = stay the same or increase, 0 = decrease) were introduced as proxy variables for market demand. As shown in Table 7, three findings are noteworthy. First, the lockdown directly stifled the rebound in market demand, and had a more significant negative impact on market expectations. Second, as time passed, the *shutdown and anti-epidemic effects* gradually appeared, having a significant impact on the market expectations of enterprises in June. Third, when the market demand is controlled for in column (5), the market demand

**Table 6. Logit regression of work and production resumption on lockdown and COVID-19 cases in other regions.**

| | labdummy | labdummy | resdummy | resdummy | resdummy |
|---|---|---|---|---|---|
| main | | | | | |
| rr_m | -0.449 | -0.511 | -0.285 | -0.414* | -0.249 |
| | (0.327) | (0.342) | (0.236) | (0.232) | (0.224) |
| otherrcv_4mn | -0.0857 | -0.218 | -0.101* | -0.307*** | -0.257*** |
| | (0.0833) | (0.134) | (0.0609) | (0.0664) | (0.0642) |
| othercfm_4mn | 0.0218 | 0.170 | 0.0259 | 0.329*** | 0.215*** |
| | (0.0221) | (0.165) | (0.0161) | (0.0647) | (0.0748) |
| Cross-regional shutdown effect | | 0.113 | | 0.167*** | 0.112*** |
| | | (0.0744) | | (0.0292) | (0.0360) |
| Cross-regional anti-epidemic effect | | -0.142 | | -0.292*** | -0.177** |
| | | (0.154) | | (0.0611) | (0.0736) |
| rcv_4mn | -0.0689 | -0.0263 | -0.0737 | 0.0143 | -0.0401 |
| | (0.0834) | (0.0996) | (0.0605) | (0.0646) | (0.0615) |
| cfm_4mn | -0.0218 | -0.0551 | -0.0375 | -0.236*** | -0.131** |
| | (0.0540) | (0.158) | (0.0570) | (0.0473) | (0.0556) |
| labret | | | | | 1.412*** |
| | | | | | (0.0307) |
| pseudo $R^2$ | 0.100 | 0.100 | 0.085 | 0.086 | 0.247 |
| N | 44,061 | 44,061 | 44,072 | 44,072 | 44,072 |

Note: The sample is the mixed cross-sectional data of enterprises interviewed in the second and third questionnaire surveys. The dependent variables are dummies for whether production and work are basically resumed. The independent variables are the average emergency response level of each city and number of recoveries and infections in other regions within the same province in April. Models 2 and 4 include the interaction between COVID-19 cases (number of recoveries and infections in April) in other regions within the same province and lockdown in addition to the variables in Models 1 and 3, respectively. Model 5 includes the work resumption level in addition to the variables in Model 2. Other control variables and fixed effects are the same as in Table 3. Numbers in brackets are standard errors clustered at the city level.

* $p < 0.1$
** $p < 0.05$
*** $p < 0.01$

significantly promoted enterprises' resumption of production, and both the *shutdown and anti-epidemic effects* were reduced (compared with column (5) of **Table 4** and column (5) of **Table 7**). This indicates that market demand, especially enterprises' subjective market expectations, is another channel through which containment measures or the lockdown affects the resumption of production and generates social costs.

To summarize, the baseline regression results show that containment measures, such as lockdowns, primarily have shutdown and anti-epidemic effects on enterprise production resumption. Approximately half of such effects are achieved via three channels: labor input (resumption of work), supply chain movement (cross-regional influence), and market demand (enterprises' market expectations).

## Moderation of the shutdown and anti-epidemic effects

The regression results of the empirical model (2a) are presented in **Table 8**, with Panel A reporting the moderation of the *shutdown effect* and Panel B reporting the moderation of the *anti-epidemic effect*. As shown in M1-M6, proxy variables of testing and quarantine resources (proportion of physicians and average hospital size), government resource mobilization capacity (proportion of government expenditure and proportion of employees in urban non-private

**Table 7. Impact of containment measures on orders and market expectations.**

| | ordup | ordup | dmdexp2 | dmdexp2 | resdummy |
|---|---|---|---|---|---|
| | May | June | May | June | Mixed |
| rr_m | -0.116 | -0.525** | -0.401* | -0.581** | -0.128 |
| | (0.160) | (0.212) | (0.210) | (0.229) | (0.189) |
| rcv_4mn | -0.00736 | 0.0752 | 0.0219 | 0.153** | 0.188*** |
| | (0.0302) | (0.0545) | (0.0414) | (0.0690) | (0.0545) |
| cfm_4mn | 0.0617 | -0.0546 | 0.0244 | -0.297** | -0.299*** |
| | (0.0570) | (0.108) | (0.0820) | (0.138) | (0.105) |
| Shutdown effect | 0.0129 | -0.0393 | -0.00251 | -0.0885** | -0.0975*** |
| | (0.0210) | (0.0315) | (0.0231) | (0.0356) | (0.0357) |
| Anti-epidemic effect | -0.0504 | 0.0307 | -0.0295 | 0.173** | 0.153** |
| | (0.0445) | (0.0646) | (0.0485) | (0.0728) | (0.0728) |
| labret | | | | | 1.372*** |
| | | | | | (0.0313) |
| ordup | | | | | 0.443*** |
| | | | | | (0.0339) |
| dmdexp1 | | | | | 0.264*** |
| | | | | | (0.0207) |
| 1.foreign | -0.00888 | -0.144*** | -0.144*** | -0.280*** | -0.00569 |
| | (0.0441) | (0.0505) | (0.0389) | (0.0468) | (0.0378) |
| 1.capsat | 0.221*** | 0.463*** | 0.157*** | 0.331*** | 0.170*** |
| | (0.0385) | (0.0379) | (0.0356) | (0.0380) | (0.0391) |
| pseudo $R^2$ | 0.026 | 0.040 | 0.021 | 0.029 | 0.263 |
| N | 20,630 | 22,087 | 21,190 | 22,878 | 42,717 |

Note: The samples are the cross-sectional data of enterprises interviewed in the second and third questionnaire surveys, respectively, as well as the mixed data. The dependent variables in columns (1)-(2) are month-on-month changes in order quantity. The dependent variables in columns (3)-(4) are enterprises' market expectations. The dependent variable in column (5) is the dummy for whether production is basically resumed. The independent variables related to the epidemic and its prevention include the average emergency response level *rr_m*, number of recoveries *rcv_4mn*, and number of infections *cfm_4mn* in each city in April, as well as the interaction of *rr_m* with *rcv_4mn* and *cfm_4mn* (to estimate the shutdown and anti-epidemic effects, respectively). Model 5 includes the work resumption level, changes in order quantity, and enterprises' market demand expectations in addition to the variables in Model 2. Other control variables and fixed effects are the same as in Table 3. Numbers in brackets are standard errors clustered at the city level.

\* $p < 0.1$

\*\* $p < 0.05$

\*\*\* $p < 0.01$.

units to the population), and government attitude towards epidemic containment (foreign trade dependence and external financing level) are used as the moderating variables of the *shutdown effect*. The interaction between the digitization level (proportion of employees in the ICT industry to the population) and the *anti-epidemic effect* is controlled. Two findings are noteworthy. First, the quantity and quality of testing and quarantine resources and the government's ability to mobilize resources positively moderated the *shutdown effect*, lowering the output cost due to the *shutdown effect*. Second, the coefficients of the moderating effects of foreign trade dependence and external financing level on the shutdown effect are also significantly positive. This is consistent with Hypothesis 2b and implies that the government's preference for scientific and targeted response mitigates the shutdown effect, whereas aggressive containment measures, such as one-size-fits-all response, aggravate the shutdown effect. In M1 of Panel A and M7-M10 of Panel B, the local digitalization level (proportion of employees

**Table 8. Moderating effects on the relationship between containment measures and production resumption.**

| Panel A | Moderation of shutdown effect | | | | | |
|---|---|---|---|---|---|---|
| Model | M1 | M2 | M3 | M4 | M5 | M6 |
| Shutdown effect moderating variable | Proportion of physicians | Average hospital size | Proportion of government expenditure | Proportion of employees in urban non-private units | Foreign trade dependence | External financing level |
| Anti-epidemic effect moderating variable | Proportion of ICT employees | | | | | |
| rr_m | -0.239 | -0.210 | -0.188 | -0.192 | -0.187 | -0.318** |
| | (0.190) | (0.173) | (0.169) | (0.178) | (0.179) | (0.145) |
| Shutdown effect | -0.237*** | -0.378*** | -0.253*** | -0.121*** | -0.134*** | -0.0876** |
| | (0.0346) | (0.115) | (0.0555) | (0.0355) | (0.0328) | (0.0347) |
| Anti-epidemic effect | 0.163*** | -0.0290 | 0.386*** | 0.252*** | 0.255*** | 0.170*** |
| | (0.0515) | (0.0822) | (0.0901) | (0.0642) | (0.0601) | (0.0653) |
| Shutdown effect moderating variable # shutdown effect | 0.00300*** | 0.00137*** | 0.996*** | 0.00000754*** | 0.0646*** | 12.78*** |
| | (0.000464) | (0.000503) | (0.278) | (0.00000101) | (0.00884) | (4.106) |
| Anti-epidemic effect moderating variable # anti-epidemic effect | -0.000776*** | -0.000996*** | -0.000491*** | -0.000461*** | -0.000418*** | 0.0000762 |
| | (0.000121) | (0.000372) | (0.000142) | (0.0000807) | (0.0000724) | (0.0000501) |
| labret | 1.412*** | 1.412*** | 1.415*** | 1.414*** | 1.417*** | 1.422*** |
| | (0.0310) | (0.0310) | (0.0309) | (0.0308) | (0.0312) | (0.0318) |
| pseudo $R^2$ | 0.246 | 0.246 | 0.247 | 0.247 | 0.247 | 0.248 |
| N | 43500 | 43500 | 43979 | 43979 | 43457 | 42035 |
| Panel B | Moderation of anti-epidemic effect | | | | | |
| Model | M7 | M8 | M9 | M10 | M11 | M12 |
| Shutdown effect moderating variable | Proportion of physicians | | | | | |
| Anti-epidemic effect moderating variable | Number of mobile phone subscribers per capita | Weibo Government Affairs Index | Per capita postal service income | Average PM2.5 concentration | Population density | Household size |
| rr_m | -0.216 | -0.214 | -0.221 | 0.399** | -0.219 | -0.276 |
| | (0.195) | (0.191) | (0.191) | (0.187) | (0.191) | (0.197) |
| Shutdown effect | -0.181*** | -0.125*** | -0.150*** | -0.168*** | -0.108** | -0.0771 |
| | (0.0423) | (0.0450) | (0.0366) | (0.0510) | (0.0446) | (0.0501) |
| Anti-epidemic effect | 0.614*** | 0.392*** | 0.322*** | 0.817*** | 0.106 | -0.0388 |
| | (0.134) | (0.0694) | (0.0557) | (0.310) | (0.0679) | (0.132) |
| Shutdown effect moderating variable # shutdown effect | 0.00236*** | -0.0000752 | 0.00193*** | -0.00163* | 0.000308 | -0.000444 |
| | (0.000702) | (0.000236) | (0.000288) | (0.000861) | (0.000226) | (0.000428) |
| Anti-epidemic effect moderating variable # anti-epidemic effect | -0.0971*** | -0.00260*** | -0.0000341*** | -0.0138** | 0.0417*** | 0.0668* |
| | (0.0274) | (0.000410) | (0.00000502) | (0.00703) | (0.00587) | (0.0393) |
| labret | 1.412*** | 1.412*** | 1.411*** | 1.395*** | 1.412*** | 1.412*** |
| | (0.0310) | (0.0310) | (0.0310) | (0.0344) | (0.0309) | (0.0310) |
| pseudo R2 | 0.246 | 0.246 | 0.247 | 0.243 | 0.246 | 0.246 |

(*Continued*)

**Table 8.** (Continued)

| Panel A | Moderation of shutdown effect | | | | | |
|---|---|---|---|---|---|---|
| N | 43,593 | 43,593 | 43,519 | 36,100 | 43,593 | 43,593 |

Note: The dependent variable is the dummy for whether production is basically resumed. The common independent variables of all models are the average emergency response level in April, its interaction with the number of new recoveries and new cases in April, and control variables (work resumption level, enterprise characteristics, such as size, industry fixed effect, and provincial fixed effect). A triple interaction between moderating variables of the shutdown effect (proportion of physicians to the population, average hospital size (number of doctors/ number of hospitals), proportion of government expenditure in GRP, and proportion of employees in urban non-private units to the population to measure testing and quarantine resources and government resource mobilization capacity; foreign trade dependence and external financing level to measure the government's tendency to implement aggressive containment policies), the emergency response level, and the number of new recoveries, and a triple interaction between moderating variables of the anti-epidemic effect (proportion of ICT employees to the population, number of mobile phone subscribers per capita, Weibo Government Affairs Index, per capita postal service income, and PM2.5 to measure the lockdown effectiveness; population density and household size to measure the virus transmission rate), the emergency response level, and the number of new cases are included in different models. Different moderating variables of the shutdown effect are included in M1-M6. Different moderating variables of the anti-epidemic effect are included in M7-M12. Numbers in brackets are standard errors clustered at the city level.

$* \ p < 0.1$

$** \ p < 0.05$

$*** \ p < 0.01.$

in the ICT industry to the population, number of mobile phone subscribers per capita, and Weibo Government Affairs Index) and external environment (per capita postal service income and average PM2.5 concentration) are used as the proxy variables for lockdown effectiveness, and control for the interaction between medical resources (proportion of licensed physicians to the population) and the shutdown effect. The results show that, for the anti-epidemic effect, the coefficients of the moderating effects are all significantly negative, indicating that a further increase in the lockdown effectiveness will marginally reduce the *anti-epidemic effect* of the lockdown. According to Hypothesis 2b and the theoretical model (1a), when the lockdown effectiveness $\theta > \theta^* = 1/(2L)$, its further increase will lead to a marginal reduction in the *anti-epidemic effect* due to the reduced possibility of virus transmission from the lockdown areas and fewer new infections. Conversely, when the lockdown effectiveness $\theta \leq \theta^*$, the anti-epidemic effect of the lockdown will be marginally increased with the lockdown effectiveness. Since the surveys were conducted in May and June 2020, when COVID-19 was well under control and community containment measures were effectively implemented, the anti-epidemic effect of the lockdown already demonstrated clear diminishing marginal returns as lockdown effectiveness increased.

Finally, population density and average household size are used to measure the virus transmission rate in M11-12 of Panel B. In the early stage of the COVID-19 pandemic, as the novel coronavirus did not mutate, the basic reproduction number remained unchanged. However, in practice, the higher the population density, the larger the household size, the more intense the interpersonal contact, the easier and faster virus transmission. According to the findings, cities with a higher population density or household size experienced a stronger *anti-epidemic effect* from the lockdown. This also validates Hypothesis 2a. A wider lockdown of populous areas is therefore beneficial for the long-term revival of local economy.

## Discussion

The majority of existing literature has found that epidemic control policies have a negative economic impact, worsening the economy at the macro level and exacerbating employment and income inequality at the micro level. Dreger and Gros (2020) reported that social

distancing restrictions, as measured by the Oxford stringency indicator, had a strong correlation with economic recession and recovery, whereas the increase in new infections and deaths did not directly limit economic recovery [25]. Mandel and Veetil (2020) estimated the economic cost of the lockdown, revealing a 7% reduction in global output with the lockdown in China alone and a 23% reduction with the lockdown in many countries [26]. Fang et al. (2020) found that COVID-19 cases abroad and pandemic control policies pursued by foreign governments reduced new job creations in China by 11.7% through the global supply chain based on data from job search websites [27]. Martin et al. (2020) reported that assuming a shelter-in-place period of three months, the poverty rate would temporarily increase from 17.1% to 25.9% in the San Francisco Bay Area in the absence of unemployment insurance or federal incentives provided by the CARES Act [28]. Using individual tracking data in the Chinese labor market, Cai (2021) found a V-shaped employment pattern during the COVID-19 pandemic, and that the city lockdown significantly slowed down work resumption, causing short-term economic and psychological losses to workers [29]. This work combines the study of optimal epidemic management strategy with the study of the economic impact of Covid-19 to investigate the trade-off between immediate output loss and future economic growth for epidemic containment measures, not only adding to the literature on the economic effects of epidemics and their control measures, but also suggesting several important implications for the Covid-19 outbreak.

First of all, to speed up the return to work and production, local governments should implement stringent containment measures in the early stages of the pandemic and lift the lockdown in a timely manner. The anti-epidemic effect is in an inverted U-shaped pattern. It is large enough in the early stages of the epidemic to make the lockdown policy cost-effective. When infected individuals are quarantined and have a very low probability of coming into contact with the susceptible, the lockdown should be lifted because the cost of the shutdown outweighs the anti-epidemic benefit. However, as the epidemic spreads due to a late or ineffective lockdown, the anti-epidemic effect diminishes and even reverses, and hence the lockdown raises social costs. This also elucidates the theoretical intuition of "stringent quarantine" and "herd immunity" policies. In order to slow the spread of COVID-19 at the beginning of the epidemic, many countries implemented a number of policies, including quarantine measures as well as lockdowns or social distancing [30, 31]. As the epidemic spreads, the economic downturn or economic crisis forces governments to relax epidemic control policies. Some countries with low lockdown effectiveness have even adopted herd immunity in order to resume economic operations.

Second, in order to maximize the lockdown's net value, governments should combine lockdown with other measures to strengthen the anti-epidemic benefit while controlling the shutdown costs. As implied in the moderation analysis, local governments should implement timely city lockdowns of densely populated cities, and ensure the timely return to work of low-risk individuals by improving quarantine efficiency and vertical administrative efficiency. Sufficient mobilized testing and quarantine resources and government attitude towards epidemic control can mitigate the shutdown effect and help reduce output costs. Bonacini et al. (2021) evaluated the effectiveness of three lockdowns in Italy using machine learning [32]. They found that the first social distancing measures had a strong effect by direct restrictions and influencing public perceptions of COVID-19, but with delays, and that the two subsequent lockdowns had a reduced but faster effect due to the replenishment of medical resources and the improvement of testing procedures and technology. Similarly, using the SIR model, Atkeson et al. (2020) demonstrated that virus screening and diagnosis techniques effectively reduced the number of healthy employees being locked down, and that its economic and health benefits far outweighed its costs [33]. Furthermore, local governments should develop

effective supporting policies for epidemic control in order to boost market confidence and thus accelerate the reopening of businesses. Fiscal and social security policies, such as targeted subsidies, unemployment insurance, and financial support, not only effectively alleviate unemployment, but also shorten the time to income recovery by several months [28].

Last but not least, the government can use digital measures to substitute some of the lockdown measures. On the one hand, it is critical to improve the vertical information system between the government, the grassroots, and the public by utilizing smart digital technologies in order to address public health emergencies and economic crises. A government is motivated to use news or other forms of information to raise economic expectations and quell social unrest [34, 35]. In reality, though, grassroots organizations like township governments or community committees, have poor communication with the government at the higher level and information asymmetry with citizens at the lower level, resulting in an increase in policy intensity at every lower level. Vertical information obstruction continues to be a significant obstacle to improving the quarantine efficiency and hence reducing economic costs of the lockdown. The application of smart digital technologies in China, such as health codes, itinerary codes, and location codes, has realized two-way information transfer between government and citizens, which has greatly improved the efficiency of COVID-19 detection and follow-up of at-risk individuals. In this approach, digital measures can significantly assist the government in ensuring the implementation of the "dynamic zero-out" policy and making the response more scientific and targeted based on the changing scenario.

On the other hand, efforts should also be made to improve public compliance with containment measures by using digital tools. Digitalization enables remote work or study, as well as faster horizontal transmission of epidemic information, which may improve public compliance with containment measures [22]. Research shows that the release of information related to the epidemic and its prevention policies through the government affairs microblog platform helps enhance public compliance [23]. The use of remote supporting devices (e.g., telepsychiatry or teleconsultation) to provide the general public with continued access to primary and mental health care services during quarantine and self-isolation assists the public in adhering to the recommendations not to visit healthcare facilities [36, 37]. In regions with strong citizen compliance with staying at home due to a high digitalization level, stringent containment measures can be relaxed by strengthening the publicity and education of the epidemic through the government affairs platform, encouraging working from home, enhancing big data tracking, and implementing home monitoring with a community grid supervision and reporting system. For a virulent virus with a great risk of transmission, i.e. a low $\theta$, digital tools can improve the lockdown effectiveness and amplify the anti-epidemic effect. Therefore, the government must upgrade information systems and implement stringent lockdown measures at the same time.

The study has several limitations. The ACFIC surveys on private enterprise operation are combined to create a mixed cross-sectional dataset, which lacks the flexibility of panel data. Although we modeled lockdown and quarantine to the SIR model to exhibit greater explanatory power in terms of the economic effect of containment measures during an epidemic, it still had some inherent limitations, such as a lack of modeling agents' heterogeneity and spatial characteristics. A more efficient model that makes use of neutral networks' powerful learning ability awaits further investigation [38]. Because we measured the lower bound of the fatality cost without considering the extra cost of a death, our estimates of the "anti-epidemic effect" are somewhat understated. This estimate bias will not jeopardize our findings but lend strength to favor the inverted U-shaped pattern of the lockdown policy's effect. If the costs associated with deaths are deemed substantial or if mortality rates among children and the elderly are comparatively high, it is advisable to enforce more stringent and prolonged lockdown policies. Furthermore, while we modeled the output costs and fatality costs, we were less

likely to be able to account for other social costs of containment measures, such as fiscal pressure on local governments and harm to people's mental health. Public expenditure for COVID-19 control by governments at all levels in China was 116.9 billion yuan as of March 13 in 2020, primarily made up of the costs of testing and quarantine, treatment, COVID-19 prevention supplies, construction of treatment facilities, and accommodation and subsidies for medical staff and other front-line personnel. The lockdown policies during the pandemic have caused psychological losses to citizens [29, 39]. Suicide risk during COVID-19 pandemic is sustained and a vicious cycle may be created with the interaction among psychiatric, psychological, and social factors such as income decrease, unemployment and repaying debts difficulty [4]. Nevertheless, the containment measures have yielded some social benefits, such as improved air quality [1]. To investigate the psychological effects on people or the allocative efficiency of government expenditure, a more comprehensive model and a broader and more detailed dataset should be future explored.

## Conclusions

Using a simple SIR model, this study investigates the dynamic impact of containment measures such as lockdown on economic activity and the underlying mechanism. Following that, the current *shutdown effect* and long-term *anti-epidemic effect* of the lockdown policy, as well as their moderating factors, are validated by combining data from the ACFIC surveys on private enterprise operation in China in 2020 with COVID-19 statistics and the regional-level heterogeneity. Furthermore, three channels through which the lockdown affects reopening are identified: labor input, factor mobility, and market demand. This not only contributes to a better understanding of the rationality and effectiveness of the lockdown policy, but also provides a theoretical foundation and practical evidence for the government to effectively implement regional- or community-specific lockdowns and targeted responses in future epidemic prevention and control.

Theoretically, the restricted movement of low-risk and recovered individuals is the primary source of social costs caused by the lockdown policy. Planners should make every effort to keep such people's lives as normal as possible, and to minimize the career disruption caused by the epidemic and the lockdown policy. On the other side, the lockdown's *anti-epidemic effect* grows and then diminishes as virus spreads. Hence, planners must implement and lift the lockdown in a timely manner to minimize social costs associated with new infections.

Practically, Governments at all levels should fully mobilize quarantine and management resources, as well as develop scientific and reasonable criteria for lifting the lockdown, all while continuing to implement regular epidemic prevention and control. Additionally, the control of the outbreak can benefit from the use of digital measures. On the one hand, the government must use the vertical information system to obtain timely information about the epidemic and accurately define lockdown zones based on population movement data. On the other hand, horizontal information systems and remote supporting devices should be developed to address the contradiction between public discontent and the risk of virus transmission, to strengthen the public compliance with self-protection, health monitoring, and other local epidemic control policies, and to avoid secondary disasters caused by the epidemic's over-consumption of medical resources, such as inadequate treatment of other health conditions.

## Supporting information

**S1 Table. Descriptive statistics of COVID-19 data in prefecture-level cities in China (excluding imported or unconfirmed cases).**
(PDF)

## Acknowledgments

The authors are grateful to All-China Federation of Industry and Commerce for the supporting of investigation and data.

## Author Contributions

**Conceptualization:** Wenxuan Chen.

**Data curation:** Jianliang Ye.

**Formal analysis:** Wenxuan Chen, Songlei Chao.

**Funding acquisition:** Songlei Chao.

**Investigation:** Wenxuan Chen, Songlei Chao.

**Methodology:** Wenxuan Chen.

**Project administration:** Wenxuan Chen.

**Supervision:** Jianliang Ye.

**Validation:** Wenxuan Chen, Songlei Chao, Jianliang Ye.

**Visualization:** Wenxuan Chen.

**Writing – original draft:** Wenxuan Chen, Songlei Chao.

**Writing – review & editing:** Wenxuan Chen, Songlei Chao, Jianliang Ye.

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
