## [Decision Letter · Decision Letter 0]

9 May 2023

PONE-D-23-06635The Micro-Economic Effects of COVID-19 Containment Measures: A Simple Model and Evidence from ChinaPLOS ONE

Dear Dr. Chao,

Thank you for submitting your manuscript to PLOS ONE. After careful consideration, we feel that it has merit but does not fully meet PLOS ONE’s publication criteria as it currently stands. Therefore, we invite you to submit a revised version of the manuscript that addresses the points raised during the review process.

We look forward to receiving your revised manuscript.

Kind regards,

Jing Cheng

Academic Editor

PLOS ONE

“This research was funded by Zhejiang Provincial Philosophy and Social Sciences Planning Project, grant number 23NDJC389YBM and Zhejiang Digital Government Construction Theory Research Project of Zhejiang Provincial Big Data Development Admin-istration, grant number ZZCG2022F-CS-108.”

4. Please remove your figures from within your manuscript file, leaving only the individual TIFF/EPS image files, uploaded separately. These will be automatically included in the reviewers’ PDF.

Reviewers' comments:

Reviewer's Responses to Questions

**Comments to the Author**

1. Is the manuscript technically sound, and do the data support the conclusions?

Reviewer #1: Yes

Reviewer #2: Yes

2. Has the statistical analysis been performed appropriately and rigorously? 

Reviewer #1: I Don't Know

Reviewer #2: Yes

3. Have the authors made all data underlying the findings in their manuscript fully available?

Reviewer #1: Yes

Reviewer #2: Yes

4. Is the manuscript presented in an intelligible fashion and written in standard English?

Reviewer #1: Yes

Reviewer #2: Yes

5. Review Comments to the Author

Reviewer #1: Dear/s Author/s,

Re: Manuscript “The Micro-Economic Effects of COVID-19 Containment Measures: A Simple Model

and Evidence from China”

Reviewer’s report:

The aim of the paper is relevant to propose solutions to the conflict between economics and health in times of health crisis such as the one that has occurred with COVID-19. The introduction has allowed the objective of the study to be well stated, in view of the existing scientific literature. The methodology used, based on the SIR model, is adequate to meet these objectives. The results and practical implications obtained from the work are relevant for future crisis situations. An interesting discussion with the existing scientific literature is also observed.

Best regards

Reviewer #2: See attached report.

(1) Value of Statistical life: While in the major western countries, there exist a good body of analytical work estimating the value of a statistical life, a key ingredient in valuing the social value of anti-epidemic effects. I am not aware of any studies of this nature in the Chinese context. How did the authors get around this?

(2) Population Heterogeneity: Naturally in any context, the universe of the underlying population at risk, i.e., the susceptible, would exhibit heterogeneity in the shape of natural immunity from lifestyle/occupational hazards, exposure to prior health/viruses, and of course there are those who are immune compromised in various degrees. How does a panel data of morbidity allow analysts to model the heterogeneity in empirical work? What are the pitfalls in ignoring this element?

(3) Quality of data: The analysis undertaken here is data intensive, which implies that a similar analysis cannot be done in every context, e.g., in the context of countries with limited resources and skill shortages. In western media, perhaps unfairly, there has been doubts on the quality of Chinese epidemiological data, especially during the early COVID onslaught in Wuhan. Some mention ought to be made of what was done to cross-check /validate the data used in the analysis performed here.

(4) Testing & Quarantine: The authors report that “the proportion of licensed physicians to the urban population is used to measure the quantity of testing and quarantine resources, the average size of hospitals (number of doctors/number of hospitals) to measure the quality of testing and quarantine resources …”. A recognition ought to be made that these means may not be suitable in contexts due to the physical constraints (e.g., due to supply chain issues) on the supply of kits/essential devices had been indifferent in different countries, and often within a country between central cities and the regional centres.

6. PLOS authors have the option to publish the peer review history of their article (what does this mean?). If published, this will include your full peer review and any attached files.

Reviewer #1: No

Reviewer #2: **Yes: **Syed M. Ahsan

---

## [Author Response · Author response to Decision Letter 0]

1 Jun 2023

Responses to Academic Editor 

Thank you for bringing this to our attention. .

To ensure compliance with PLOS ONE's style requirements, we have made careful adjustments to the formatting of our manuscript. Specifically, we have reviewed and modified each formula and variable, ensuring that they meet the journal’s style requirements. Additionally, we meticulously compared and adjusted the formatting of each reference, including the citation style and bibliography format. We have also made necessary modifications to the format of our funding statement. Furthermore, we have revised and refined the presentation of our figures and tables.

Moreover, we conducted a thorough proofreading of the entire manuscript, with a focus on addressing language-related issues. We have made further improvements to enhance the overall quality of the writing and to rectify any identified language problems.

We believe that these extensive modifications, coupled with the subsequent careful review, have significantly improved the manuscript's presentation and adherence to the guidelines set by the journal.

Should you require any additional information or have further concerns, please do not hesitate to contact us. We are more than willing to promptly provide any necessary documentation or clarification to address this matter effectively.

“This research was funded by Zhejiang Provincial Philosophy and Social Sciences Planning Project, grant number 23NDJC389YBM and Zhejiang Digital Government Construction Theory Research Project of Zhejiang Provincial Big Data Development Administration, grant number ZZCG2022F-CS-108.”

If this statement is not correct you must amend it as needed. Please include this amended Role of Funder statement in your cover letter; we will change the online submission form on your behalf.

Thank you for your feedback. We have considered your suggestion and revised the funding statement in accordance with the guidelines provided by the journal and the formatting used in previously published articles. Our revised funding statement now reads as follows: “This research was funded by Zhejiang Provincial Philosophy and Social Sciences Planning Project(www.zjskw.gov.cn), under Grant Agreement No. 23NDJC389YBM and Zhejiang Provincial Big Data Development Administration’s Zhejiang Digital Government Construction Theory Research Project, under Grant Agreement No. ZZCG2022F-CS-108. The funders had no role in study design, data collection and analysis, decision to publish, or preparation of the manuscript.”

Thank you for bringing this to our attention. Regarding the editor’s concern regarding the copyright of the map in Figure 1, we would like to clarify that the image was created by the authors themselves, and it does not incorporate any copyrighted maps or proprietary data from external sources, such as Google Maps, Google Earth, or similar platforms. Therefore, we believe that there is no requirement for obtaining permission from any original copyright holder. 

To address this issue, we would like to provide the following information and assurances: 

The map presented in Figure 1 was entirely created by the authors using open-access and publicly available geographic information sources. It does not contain any copyrighted elements or proprietary data. To further support our claim, we have uploaded the original images created using Stata and data sources to the Open Science Framework (OSF) repository at https://osf.io/5p8hy/. These files will provide a clear demonstration of our independent creation of the figures and alleviate any concerns regarding copyright infringement. 

We understand the importance of properly attributing sources and complying with copyright guidelines. In the revised manuscript, we will clearly indicate that the map in Figure 1 was created by the authors, and appropriate citations will be provided for any data sources used in its creation. 

We affirm that the entire manuscript, including the images, will be published under the Creative Commons Attribution License (CC BY 4.0), ensuring free access, distribution, and use of the material with proper attribution. 

Given these considerations, we kindly request the reviewers and the editorial team to reconsider their concerns regarding the copyright of Figure 1. We believe that our explanation demonstrates our adherence to the copyright guidelines outlined by PLOS and ensures that our research respects the principles of open access and proper attribution. 

If you require any further information or have any additional concerns, please do not hesitate to contact us. We are more than willing to provide any necessary documentation or clarification to address this matter promptly. 

4. Please remove your figures from within your manuscript file, leaving only the individual TIFF/EPS image files, uploaded separately. These will be automatically included in the reviewers’ PDF.

Thank you for your suggestion. We have revised the manuscript as per your recommendation by removing the figures from the manuscript file. Instead, we have uploaded the individual TIFF image files separately during the submission process.

Thank you for your request. We have carefully reviewed our reference list and made the necessary adjustments to ensure its completeness and accuracy. 

First, we have diligently revised the reference list to comply with the journal's formatting guidelines, ensuring the accuracy and completeness of each citation. 

Second, we have also meticulously checked the publication and retraction status of each reference and found no instances of retracted articles in our list.

Third, we have updated the publication status of the references to reflect the most recent information available.

Furthermore, in response to reviewer’s feedback, we have also added several additional references to further support our arguments. The specific details of these additional references are as follows:

14. Hall RE, Jones CI, Kleneow PJ. Trading off consumption and COVID-19 deaths. 2020;42(1):1-14. Quarterly Review. doi: org/10.21034/qr.4211

15. Mena GE, Martinez PP, Mahmud AS, Marquet PA, Buckee CO, Santillana M. Socioeconomic status determines COVID-19 incidence and related mortality in Santiago, Chile. Science. 2021; 372(6545):eabg5298. doi: 10.1126/science.abg5298.

16 Acemoglu D, Chernozhukov V, Werning I, Whinston MD. A multirisk SIR model with optimally targeted lockdown. NBER working paper. 2020; 1-57. doi: 10.3386/w27102.

21. Lu FS, Nguyen AT, Link NB, Molina M, Davis JT, Chinazzi M, et al. Estimating the cumulative incidence of COVID-19 in the United States using influenza surveillance, virologic testing, and mortality data: four complementary approaches. PLoS Comput Biol 17(6): e1008994. doi: org/10.1371/journal.pcbi.1008994.

We have made sure to include proper citations and full references for these newly added sources, providing transparent and accurate support for our study.

Responses to Reviewer 1

Reviewer’s report:

The aim of the paper is relevant to propose solutions to the conflict between economics and health in times of health crisis such as the one that has occurred with COVID-19. The introduction has allowed the objective of the study to be well stated, in view of the existing scientific literature. The methodology used, based on the SIR model, is adequate to meet these objectives. The results and practical implications obtained from the work are relevant for future crisis situations. An interesting discussion with the existing scientific literature is also observed.

Best regards

Thank you for your review of our paper. We greatly appreciate your positive feedback and acknowledgment of the relevance of our study in proposing solutions to the conflict between economics and health during health crises. 

We are encouraged by your recognition of the significance of the results and their practical implications for future crisis situations. Additionally, we are glad that you found our discussion engaging and relevant, considering the existing scientific literature.

Thank you once again for your valuable input and encouraging remarks.

Responses to Reviewer 2

General Remarks

This paper wishes to disentangle the negative economic impact of the pandemic lockdown policy, dubbing it ‘the shutdown effect’ (SDE) from the presumptive benefits, the ‘antiepidemic effect’ (AEE) in the context of the COVID containment measures in China. The authors make use of the ‘Susceptible-Infected-Recovered/Removed’ (SIR) model from the epidemiology literature to examine the trade-off between the two consequences of the lockdown policy in the anticipation that one can determine when it may be optimal to lift the lockdown policy. Ideally this would require the determination of the critical point where the social value of the shutdown losses matches the social value of AEE. 

The SIR model is an analytic device that is typically employed to solve an optimal control problem, which has been analysed by many, but the present authors rely on a recent paper by Alvarez et al (2021). However, instead of cross-country analysis, they model the COVID situation in the context of China with micro data (enterprise level). Morbidity data came a panel of prefecture level cities. 

The analysis seems comprehensive, and the authors generate valuable policy suggestions on the timing of the enforcement and withdrawal of the lock down policy, on the scope of testing and related preventive/mitigating public health measures as well as social interventions in ensuring the sustenance of daily existence. They also explore the scope of digital interventions where physical contacts and interventions are not practicable. 

Recommendation

I recommend publication of the paper after suitable revisions as outlined below under the Specific Comments. These are mainly in the nature of conceptual issues that need to be clarified without adding to the length of the final paper from the current length. 

Thank you for your thorough review and valuable feedback. We sincerely appreciate your recognition of our research and the positive acknowledgment of our policy suggestions.

Based on your review, we understand that there are conceptual issues that need clarification. We will carefully address these concerns in our revisions to ensure greater clarity and coherence. In our upcoming response, we will address each of your specific points and provide a comprehensive explanation.

Once again, we sincerely appreciate your positive recommendation for the publication of our paper with suitable revisions. Your feedback has greatly contributed to improving the overall quality of our research.

Thank you for your time and valuable insights.

Specific Comments

(1) Value of Statistical life: While in the major western countries, there exist a good body of analytical work estimating the value of a statistical life, a key ingredient in valuing the social value of anti-epidemic effects. I am not aware of any studies of this nature in the Chinese context. How did the authors get around this?

We sincerely appreciate your comment regarding how we address the value of a statistical life (VSL) in our model. We have found some similar studies on VSL for China, which we would like to highlight. Wang and He (2010) estimated a VSL in China to be 60 times the average household annual income. Additionally, in a closely related literature, Hall et al. (2020) used a utilitarian criterion to value the extra years of life lost due to infection and obtained a cost of about 60 times per capita annual consumption or 40 times annual GDP per capita.

In our theoretical framework, the fatality costs, the second component of social costs of the planner,Iϕ[w/r+x] is the product of the number of deaths (Iϕ) per period times the economic costs/shadow value assigned to each death ([w/r+x]), or the value of a statistical life (VSL) (revised in Line 100). There are two types of VSL estimation methods. One type measures willingness to pay or accept changes in human mortality risk, while the other type measures the present value of the loss of direct income (w/r) of the deceased without taking into consideration the extra cost (x) on individual and family well-being such as suffering, therefore is often regarded as the lower bound of the fatality cost. In our empirical analysis, we used the latter method considering data availability. Therefore, we do not rely on a specific estimate of VSL to measure the social costs but rather use an income- or production-related measure, i.e. the production resumption levels, to represent both the output cost due to the lockdown and the fatality cost, which is the loss of direct income due to deaths. According to equation (1a), the marginal fatality cost with respect to the lockdown level enters the “anti-epidemic effect” through VI, the marginal social cost (ceteris paribus) caused by a new infection.

We recognize that our approach may underestimate the "anti-epidemic effect" since it does not fully account for the extra costs associated with deaths, such as suffering. However, this bias does not compromise our findings and actually strengthens the support for the inverted U-shaped pattern of the lockdown policy's effect. If the costs associated with deaths are deemed substantial or if mortality rates among children and the elderly are comparatively high, it is advisable to enforce more stringent and prolonged lockdown policies. (newly added in Discussion in Lines 650 to 654 ) 

In conclusion, while our approach does not require a specific estimate of VSL, we have accounted for the fatality costs and consider the lower bound of the VSL in our analysis. We appreciate your valuable input, and we believe that our findings remain robust.

Thank you once again for your thoughtful comments and suggestions.

(2) Population Heterogeneity: Naturally in any context, the universe of the underlying 

population at risk, i.e., the susceptible, would exhibit heterogeneity in the shape of natural immunity from lifestyle/occupational hazards, exposure to prior health/viruses, and of course there are those who are immune compromised in various degrees. How does a panel data of morbidity allow analysts to model the heterogeneity in empirical work? What are the pitfalls in ignoring this element?

Thank you for raising an important point regarding population heterogeneity and its implications for our empirical work. We appreciate your insightful comment and would like to address it in detail.

First, our focus is primarily on heterogeneity at the prefecture level rather than at the individual level. Our underlying theoretical model assumes no heterogeneity in fatality rate (ϕ) or in diffusion/transmission rates (β). This is consistent with the policy implemented in prefecture-level cities in China during the early stages of the outbreak, where the lockdown policy was not differentiated across different types of individuals. Our model incorporates the effectiveness of the lockdown in reducing the diffusion of the virus and considers the capabilities of testing and quarantine measures to analyze heterogeneity at the regional level (Lines 110 to 116). In this context, the average fatality rate and average diffusion rate within a city are sufficient for analyzing the social costs of the lockdown policy. By utilizing a prefecture-level panel of morbidity data, which includes variables such as new cases (recoveries, deaths) and cumulative cases (recoveries, deaths), we are able to model the average fatality rate and average diffusion rate within a city, as well as capture the heterogeneity in these rates across different cities.

To account for heterogeneity, we merge the morbidity panel data with a mixed cross-sectional dataset at the enterprise level and additional data on city characteristics. This enables us to control for factors such as occupational hazards (firm size/policy or financial supports, industry fixed effect), exposure to the virus (prefecture-level population density/morbidity), and other socio-economic conditions (GRP per capita/provincial fixed effect). By doing so, we are able to estimate the average shutdown effect and the average anti-epidemic effect across cities.

Furthermore, we introduced a variety of prefecture-level moderating factors in our empirical model to examine their interactions with the shutdown effect (SDE) and the anti-epidemic effect (AEE). These moderating factors include transmission rate/exposure to viruses (proxied by population density and average household size), outdoor activity tendency/lifestyle (proxied by PM2.5 concentration and postal service income), and medical condition/possibility of testing and quarantine (proxied by the proportion of licensed physicians). This moderation analysis provides valuable insights into how the impact of the lockdown policy is differentiated across different types of cities.

We recognized the limitations of our work (previously stated in Discussion Line 648), particularly the lack of modeling individual-level heterogeneity due to the unavailability of individual-level data. We acknowledge the varying risks among different subpopulations, as well as their potential interactions with other subgroups at varying rates. This highlights the need for a "network version" of the fundamental SIR (Susceptible-Infectious-Recovered) model, enabling the analysis of group-specific or age-specific lockdowns and associated policies (Acemoglu et al., 2020). Nevertheless, we gather a comprehensive set of features pertaining to the epidemic at the prefecture level. In this way, our research effectively uncovers regional-level heterogeneity and offers suggestions for implementing region-specific lockdowns.

Thank you once again for your valuable feedback, which has greatly contributed to the clarification of our study.

(3) Quality of data: The analysis undertaken here is data intensive, which implies that a similar analysis cannot be done in every context, e.g., in the context of countries with limited resources and skill shortages. In western media, perhaps unfairly, there has been doubts on the quality of Chinese epidemiological data, especially during the early COVID onslaught in Wuhan. Some mention ought to be made of what was done to cross-check /validate the data used in the analysis performed here. 

Thank you for your recommendation regarding clarifying the quality of data used in our analysis. We would like to address this concern in our response.

First, as part of the emergency response, China established a comprehensive reporting system and made morbidity data publicly available. The daily case count of COVID-19 in Chinese cities was obtained from the Chinese Center for Disease Control and Prevention and provincial health commissions. These sources reported the cumulative confirmed, dead, and recovered COVID-19 cases in each city on a daily basis (Lines 225 to 227). Please note that the official government websites for COVID-19 updates are no longer accessible. The reason for this is that at the end of 2022, the Chinese government discontinued its epidemic control measures. On December 25, 2022, the National Health Commission announced that daily COVID-19 updates would no longer be issued. 

To ensure the reliability of our data, we cross-checked it with three other open data sources: https://www.tianditu.gov.cn/coronavirusmap, and https://github.com/BlankerL/DXY-COVID-19-Data, and https://github.com/CSSEGISandData/COVID-19 (operated by the Johns Hopkins University Center for Systems Science and Engineering). We found some inconsistencies primarily in the numbers reported for cities in Hubei province, where Wuhan is located. These numbers underwent significant adjustments officially following the Wuhan lockdown. After excluding Hubei province, we did not find any significant systematic bias between the different data sources. Besides, the reopening of Hubei province was overseen by the central government. Therefore, we made the decision to exclude any cities from Hubei province from our sample. (Lines 231 to 238)

We also recognize that the reported confirmed cases may be biased due to population-level health-seeking behavior, surveillance capacity and the presence of asymptomatic individuals across regions (Lines 310 to 315). As cited in our manuscript (Lu et al. 2020; Mena et al., 2021), such biases are well recognized. However, it is worth noting that the use of smart digital technologies in China, such as health codes, itinerary codes, and location codes, has significantly improved the efficiency of COVID-19 detection and the follow-up of at-risk individuals, especially when there were far fewer cases in March and April of 2020. These efforts have greatly reduced potential biases in the reported newly confirmed in our sample.

In summary, we excluded cities in Hubei province from our sample due to inconsistencies and major adjustments in the reported numbers following the Wuhan lockdown. We also acknowledge the improvements made in data collection and surveillance through the use of smart digital technologies in China. These measures have enhanced the accuracy and reliability of the data used in our analysis.

Thank you once again for your insightful comments, which have contributed to the refinement of our research.

(4) Testing & Quarantine: The authors report that “the proportion of licensed physicians to the urban population is used to measure the quantity of testing and quarantine resources, the average size of hospitals (number of doctors/number of hospitals) to measure the quality of testing and quarantine resources …”. A recognition ought to be made that these means may not be suitable in contexts due to the physical constraints (e.g., due to supply chain issues) on the supply of kits/essential devices had been indifferent in different countries, and often within a country between central cities and the regional centres. 

Thank you for bringing this concern to our attention.

In our empirical analysis, we took into account the potential influence of physical constraints. Specifically, we examined the interaction effect between the physical constraint, represented by the "area of city paved roads at year-end (10000 sq.m)," and the shutdown effect. However, the coefficient of this interaction term was found to be statistically insignificant (coef=-2.24×10-6; std.err.=1.79×10-6). This suggests that the physical constraint resulting from transportation infrastructure may not be a relevant indicator of the adequacy of testing and quarantine measures during the period under study. This finding aligns with the fact that China's well-developed transportation network has reduced logistical disparities between cities. With extensive coverage of high-speed rail and a vast road network, transportation in China facilitates convenient access to most part of the country. According to the information released by the National Railway Administration, by the end of 2021, the railway network in China has extended to approximately 81% of the country's counties, ensuring accessibility to a large portion of the population. In terms of high-speed rail, it has reached 93% of cities with a population of over 500,000 (Data source: https://www.chinanews.com.cn/cj/2022/06-10/9776518.shtml). In terms of the road network, the total length of highways in China is approximately 5.2 million kilometers, with high-speed expressways covering around 161,000 kilometers. It is worth noting that these highways cover nearly 100% of cities with a population of over 200,000 (Data source: https://politics.gmw.cn/2021-02/14/content_34617915.htm). Even in rural areas, all eligible villages have achieved 100% coverage in terms of passenger transportation (Data source: http://news.china.com.cn/2022-07/29/content_78347177.html). These impressive figures demonstrate the scale and reach of China's transportation network, which plays a pivotal role in ensuring efficient logistics operations and minimizing barriers to the flow of goods and resources. Furthermore, as the COVID-19 situation in China was temporarily brought under control after March 2020, the availability of testing kits and devices was ensured. 

However, the testing, diagnosis, and treatment of the large number of susceptible individuals and infected patients heavily relied on the healthcare system, including doctors and hospitals. Therefore, in this context, we believe that metrics such as "proportion of licensed physicians" and "average size of hospitals" are more appropriate for assessing the quantity and quality of testing and quarantine resources in our research. 

We have provided the necessary explanations in the Moderation Analysis section (Lines 360 to 365) as you recommended:

“We recognize that the suitability of these measures may vary in different contexts due to physical constraints, such as supply chain issues, impacting the availability of testing kits and essential devices across regions. However, China's well-developed transportation network has reduced the logistics disparity across prefecture-level cities, and, more importantly, the supply of kits/devices was mostly guaranteed as the COVID-19 was under control during the sample period. Therefore, medical conditions, including the availability of medical services and designated hospitals for testing and treatment, serve as better predictors of testing and quarantine resources.”

Composition, Grammar and Style Issues:

(i) References: The authors should follow a standard reference format, e.g., U Chicago guidelines applicable to published and unpublished/web material. 

We appreciate your comment. We have reviewed the guidelines for references provided by the journal, specifically the PLOS ONE guidelines for References (available at https://journals.plos.org/plosone/s/submission-guidelines#loc-references), and have made the necessary modifications to ensure compliance with the standard reference format.

We have carefully revised each reference in the manuscript to adhere to the required format. Additionally, we have included any newly added references that were mentioned in our response.

Thank you for highlighting this issue, and we have made the necessary updates to ensure the accuracy and conformity of our reference list.

(ii) Grammar: This is a well written paper.

Thank you for your positive feedback on the grammar and writing quality of our paper. Your kind words are encouraging, and we are grateful for your acknowledgment. We remain committed to maintaining a high standard of writing and presentation in our research work.

Note:

Wang H, He J. The value of statistical life: a contingent investigation in China. The World Bank Policy Research Working Paper. 2010;1-37. Available from https://ideas.repec.org/p/wbk/wbrwps/5421.html

Hall RE, Jones CI, Kleneow PJ. Trading off consumption and COVID-19 deaths. 2020;42(1):1-14. Quarterly Review. doi: org/10.21034/qr.4211

Acemoglu D, Chernozhukov V, Werning I, Whinston MD. A multirisk SIR model with optimally targeted lockdown. NBER working paper. 2020; 1-57. doi: 10.3386/w27102.

Lu FS, Nguyen AT, Link NB, Molina M, Davis JT, Chinazzi M, et al. Estimating the cumulative incidence of COVID-19 in the United States using influenza surveillance, virologic testing, and mortality data: four complementary approaches. PLoS Comput Biol 17(6): e1008994. doi: org/10.1371/journal.pcbi.1008994.

Mena GE, Martinez PP, Mahmud AS, Marquet PA, Buckee CO, Santillana M. Socioeconomic status determines COVID-19 incidence and related mortality in Santiago, Chile. Science. 2021; 372(6545):eabg5298. doi: 10.1126/science.abg5298.

---

## [Decision Letter · Decision Letter 1]

2 Jul 2023

The Micro-Economic Effects of COVID-19 Containment Measures: A Simple Model and Evidence from China

PONE-D-23-06635R1

Dear Dr. Chao,

We’re pleased to inform you that your manuscript has been judged scientifically suitable for publication and will be formally accepted for publication once it meets all outstanding technical requirements.

Kind regards,

Jing Cheng

Academic Editor

PLOS ONE

Additional Editor Comments (optional):

Reviewers' comments:

Reviewer's Responses to Questions

**Comments to the Author**

1. If the authors have adequately addressed your comments raised in a previous round of review and you feel that this manuscript is now acceptable for publication, you may indicate that here to bypass the “Comments to the Author” section, enter your conflict of interest statement in the “Confidential to Editor” section, and submit your "Accept" recommendation.

Reviewer #1: (No Response)

Reviewer #2: All comments have been addressed

2. Is the manuscript technically sound, and do the data support the conclusions?

Reviewer #1: Yes

Reviewer #2: Yes

3. Has the statistical analysis been performed appropriately and rigorously? 

Reviewer #1: I Don't Know

Reviewer #2: Yes

4. Have the authors made all data underlying the findings in their manuscript fully available?

Reviewer #1: Yes

Reviewer #2: Yes

5. Is the manuscript presented in an intelligible fashion and written in standard English?

Reviewer #1: Yes

Reviewer #2: Yes

6. Review Comments to the Author

Reviewer #1: My opinion regarding this paper was in favor of its publication. I see no changes in the submission, so I continue to be in favor of it. If I need to revise something in particular, please let me know.

Reviewer #2: The authors have taken all my comments with a great deal of seriousness and clarified many points in detail that had, in part, been implicit in the previous version. They have also added some new clarifications in the text by way of responding to the responses to some of the queries I had raised. I am fully satisfied with the new version of the paper.

7. PLOS authors have the option to publish the peer review history of their article (what does this mean?). If published, this will include your full peer review and any attached files.

Reviewer #1: No

Reviewer #2: **Yes: **Professor Syed M. Ahsan

---

## [Editor Report · Acceptance letter]

7 Jul 2023

PONE-D-23-06635R1 

The Micro-Economic Effects of COVID-19 Containment Measures: A Simple Model and Evidence from China 

Dear Dr. Chao:

I'm pleased to inform you that your manuscript has been deemed suitable for publication in PLOS ONE. Congratulations! Your manuscript is now with our production department. 

Kind regards, 

on behalf of

Dr. Jing Cheng 

Academic Editor

PLOS ONE